# GEQ: Gaussian Kernel Inspired Equilibrium Models

**Mingjie Li[1], Yisen Wang[1,2], Zhouchen Lin[1,2,3]** *
[1] National Key Lab of General Artificial Intelligence,
School of Intelligence Science and Technology, Peking University
[2] Institute for Artificial Intelligence, Peking University
[3] Peng Cheng Laboratory
lmjat0111@outlook.com, {yisen.wang, zlin}@pku.edu.cn

## Abstract

Despite the connection established by optimization-induced deep equilibrium models (OptEqs) between their output and the underlying hidden optimization problems, the performance of it along with its related works is still not good enough especially when compared to deep networks. One key factor responsible for this performance limitation is the use of linear kernels to extract features in these models. To address this issue, we propose a novel approach by replacing its linear kernel with a new function that can readily capture nonlinear feature dependencies in the input data. Drawing inspiration from classical machine learning algorithms, we introduce Gaussian kernels as the alternative function and then propose our new equilibrium model, which we refer to as GEQ. By leveraging Gaussian kernels, GEQ can effectively extract the nonlinear information embedded within the input features, surpassing the performance of the original OptEqs. Moreover, GEQ can be perceived as a weight-tied neural network with infinite width and depth. GEQ also enjoys better theoretical properties and improved overall performance. Additionally, our GEQ exhibits enhanced stability when confronted with various samples. We further substantiate the effectiveness and stability of GEQ through a series of comprehensive experiments.

## 1 Introduction

Deep Neural Networks (DNNs) show impressive performance in many real-world tasks on various data like graphs [43], images [41, 20], sequences [8], and others. However, most neural network structures are constructed by experience or searching on the surrogate datasets [31, 57]. Therefore, these architectures cannot be interpretable and such a phenomenon hinders further development. Apart from the current neural network models, traditional machine learning methods like dictionary learning [46, 33], subspace clustering [54] and other methods [29, 56, 55, 30] can design their whole procedure by designing optimization problems with specific regularizers customized from their mathematical modeling and requirements. Thus, these models are easily interpreted. However, the traditional machine learning algorithms' whole procedures do not consider the hidden properties of features and labels. Therefore, they usually perform worse on tasks with more data.

To link two types of models, OptEqs [52] tries to recover the model's hidden optimization problem to make their model "mathematically explainable". They claim that the output features $\tilde{\mathbf{z}}^*$ (we also called the equilibrium state) with respect to input $\mathbf{x}$, which are obtained by solving the fixed-point

---

*Corresponding author

equation in Eqn (1), is the optimal solution for its hidden optimization problem defined in Eqn (2).

$$\tilde{\mathbf{z}}^* = \tilde{\mathbf{W}}^\top \sigma(\tilde{\mathbf{W}}\tilde{\mathbf{z}}^* + \mathbf{U}\mathbf{x} + \mathbf{b}), \quad \hat{\mathbf{y}} = \tilde{\mathbf{W}}_c \tilde{\mathbf{z}}^* + \mathbf{b}_c, \tag{1}$$

$$\min_{\tilde{\mathbf{z}}} G(\tilde{\mathbf{z}}; \mathbf{x}) = \min_{\tilde{\mathbf{z}}} \left[ \mathbf{1}^\top f(\tilde{\mathbf{W}}^{-1\top}\tilde{\mathbf{z}}) - \left\langle \mathbf{U}\mathbf{x} + \mathbf{b}, \tilde{\mathbf{W}}^{-1\top}\tilde{\mathbf{z}} \right\rangle + \frac{1}{2}\|\tilde{\mathbf{W}}^{-1\top}\tilde{\mathbf{z}}\|_2^2 - \frac{1}{2}\|\tilde{\mathbf{z}}\|_2^2 \right], \tag{2}$$

where $\sigma$ is the ReLU activation function, $\mathbf{U}, \tilde{\mathbf{W}}, \mathbf{b}, \tilde{\mathbf{W}}_c, \mathbf{b}_c$ are learnable parameters. They are trained by optimizing loss functions, such as cross-entropy loss, which are calculated based on the final prediction $\hat{\mathbf{y}}$ derived from $\tilde{\mathbf{z}}^*$ as shown in Eqn (1). $\tilde{\mathbf{W}}^{-1}$ represents an invertible or pseudo-invertible matrix for $\tilde{\mathbf{W}}$, and the function $f$ is a positive indicator function, which outputs 1 when $x \geq 0$ and outputs $\infty$ in other cases. By choosing this function, the first-order condition for Eqn 2 will contain a non-linear ReLU function, which is the activation function in our GEQ. The equivalence between $\mathbf{z}^*$ and the optimal solution for the hidden optimization problem (2) enables researchers to not only gain insights into OptEqs' behavior by understanding the underlying hidden optimization problem but also innovate by designing new models based on different problem formulations tailored to specific tasks. For instance, OptEqs introduces a module that promotes output sparsity, and Multi-branch OptEqs (MOptEqs) incorporates fusion modules to enhance the diversity among its branches. Despite these advancements and the incorporation of various modules inspired by different vision tasks, the performance of OptEqs and related models still falls short when compared to deep neural networks in image classification. This discrepancy suggests the existence of crucial components that limit the performance of equilibrium models.

To identify such a component, we delve into the hidden optimization problem of OptEqs and observe that it can be decomposed into two distinct parts: the regularizer term for the output features and the feature extraction term. While the feature extraction term is crucial as it depends on the input and determines the patterns extracted from the input features, the exploration of the regularizer term has been largely overlooked, with linear kernel functions being the predominant choice for feature extraction. Thereby, we believe that the limitations of previous equilibrium models stem from their feature extraction parts, as linear kernels struggle to capture complex features effectively. Building upon these insights, we take a step forward by leveraging the widely adopted Gaussian kernel for feature extraction in the hidden optimization problem.

Then by calculating the stationary condition for the above new hidden optimization problem, we propose our new type of OptEqs, the Gaussian kernels inspired equilibrium models (GEQ). The model involves a new attentive module induced by its hidden optimization problem and enjoys much better performances on classification tasks even compared with deep models. Furthermore, we also prove that the new model's outputs are equivalent to the outputs for OptEqs with weight-tied "infinite wide" mappings. Therefore, an interesting finding is that our model can be regarded as a "double-infinite" model because the original OptEqs can be regarded as a weight-tied "infinite deep" model. Apart from the above findings, the utilization of Gaussian kernels also makes our proposed model enjoy better generalization abilities. Besides the generalization abilities, we also analyze the stability of our GEQ and find its stability is better on various inputs. We summarize our contributions as follows:

- We first reformulate the OptEqs' hidden optimization problem with Gaussian kernels and propose a new equilibrium model called GEQ. It contains a new attention module induced by its hidden optimization problem and performs better on real-world datasets.

- We find that our GEQ can be regarded as a weight-tied neural network with both infinite width and depth, and better generalization ability through our analysis. Empirical results also confirm the superiority of our GEQ.

- We theoretically demonstrate the advantages of the stability of our GEQ compared with former OptEqs on various inputs. We also conduct experiments to validate such advantages.

## 2 Related Works

### 2.1 Implicit Models

Most modern deep learning approaches provide explicit computation graphs for forward propagations and we call these models "*explicit* models". Contrary to these models, recent researchers proposed some neural architecture with dynamic computation graphs and we call them "*implicit* models". A

notable example of an implicit model is Neural ODEs [7], its architecture is encoded as a differential system and the implicit ODE solvers they used are equivalent to continuous ResNets that take infinitesimal steps. By representing the entire structure using differential systems, implicit models tap into the black box of traditional neural networks while offering increased flexibility and interpretability. Because of the flexibility and the interpretability of implicit models, the design of implicit models [15, 17] draws much attention these days. Many kinds of implicit models have been proposed, including optimization layers [11, 1], differentiable physics engines [40, 9], logical structure learning [50], differential programming [53, 43], and others [25, 49].

Among the various implicit models, OptEqs [52] and its multi-branch version MOptEqs [28] stand out as they not only exhibit superior performance compared to other implicit models but also explicitly establish the relationship between their structure and a well-defined optimization problem. Therefore, exploring better equilibrium models is a promising direction to achieve more interpretable neural architectures. However, it is worth noting that while OptEqs and its variants have shown promising results, their performance is still not entirely satisfactory, particularly when compared to the deep explicit models. Other works [51, 42] also show the connection between their architectures and optimization problems, but their performance is also not satisfying. Besides the above general models, there are many works design equilibrium models from the view of optimization to deal with their specific tasks, like implicit graph models [6, 26], certified robust models [27] and image denoising models [5]. However, these models can only work on their specific domains.

## 2.2 Infinite Wide Models and Kernel Methods in Deep Learning

By employing kernel methods to estimate the outputs of single-layer networks for various samples, researchers discover that such networks can exhibit characteristics of a Gaussian process (GP) when their parameters are randomly initialized with a large width limit [35]. Building upon this idea, recent researchers have extended these findings to neural networks with multiple layers [24, 10] and other architectures [37, 13]. These studies primarily focus on weakly-trained models, where the network parameters are randomly initialized and kept fixed throughout the training process except for the last classification layer [2]. Despite their "weakly-trained" nature, these models still provide valuable insights applicable to current neural networks. For instance, mean-field theory [4, 16, 19] explains phenomena such as gradient vanishing and exploding during back-propagation, which are relevant not only to single-layer networks but also to other structures like convolutional neural networks (CNNs) and recurrent neural networks (RNNs). Other researchers explore stationary kernels to enhance the interpretability of neural networks by designing different activation functions [34].

In addition to weakly trained models, recent studies [22, 2] introduce the concept of Neural Tangent Kernel (NTK) and its variants. These works have demonstrated that the sample kernel of infinitely wide networks, with appropriate initialization, can converge to a fixed neural tangent kernel when trained using gradient descent with infinitesimal steps (gradient flow). The NTK model is a theoretical construct with strict constraints, and its weights are not learned. It is important to note that although our model can also be seen as an infinitely wide model, there are several key differences between our approach and the aforementioned models. Firstly, our model utilizes kernel methods to operate on input features and output features, while the NTK models employ the kernel method on samples. Secondly, our GEQ model can be viewed as employing a "weight-tied infinite wide" projection that is parameterized by learnable parameters, allowing for updates during the training process. This contrasts with NTKs and NTK-DEQ [12](an equilibrium model constructed with vanilla NTK layers), where the weights are fixed and not learned. Therefore, despite the potential overlap in terminologies used in our paper and NTK-related works, our GEQ model differs significantly.

## 3 Gaussian kernel Inspired Equilibrium models

### 3.1 Formulation and Structure of GEQ

Before starting our analysis, we need to reformulate the original formulations of OptEqs' equilibrium equation (1) and hidden optimization problem (2) for convenience. We replace $\tilde{\mathbf{W}}_c \tilde{\mathbf{W}}^\top$ with $\mathbf{W}_c$, $\mathbf{W} := \tilde{\mathbf{W}}^\top$, and replace $\mathbf{z}$ with $\tilde{\mathbf{W}}^{-1\top} \tilde{\mathbf{z}}$. Then the original OptEqs' optimization problem can be

reformulated as:

$$\min_{\mathbf{z}} G(\mathbf{z}; \mathbf{x}) = \min_{\mathbf{z}} \left[ \mathbf{1}^\top f(\mathbf{z}) + \frac{1}{2}\|\mathbf{z}\|_2^2 - \langle \mathbf{U}\mathbf{x} + \mathbf{b}, \mathbf{z} \rangle - \frac{1}{2}\|\mathbf{W}\mathbf{z}\|_2^2 \right]. \tag{3}$$

With the new formulation, we can rewrite the equilibrium equation for OptEqs with input $\mathbf{x}$ by calculating Eqn (3)'s first order stationary condition $\nabla G = \mathbf{0}$ and then reformulate it as follows:

$$\mathbf{z}^* = \sigma\left(\mathbf{W}^\top \mathbf{W}\mathbf{z}^* + \mathbf{U}\mathbf{x} + \mathbf{b}\right), \tag{4}$$

where $\sigma$ is the ReLU activation function, $\mathbf{U}, \mathbf{W}, \mathbf{b}$ are learnable parameters trained by optimizing loss functions (like cross-entropy loss). From problem Eqn (3), one can see that the GEQ's outputs try to extract features by minimizing the similarity term with the input feature $\mathbf{U}\mathbf{x} + \mathbf{b}$ through a linear kernel function with some constraints defined in its regulation terms to prevent the trivial outputs. Such an explanation can also extend to other DEQs [3, 51] under the symmetric weight constraints. However, as other traditional machine learning mechanisms show, linear kernel functions cannot perform well when processing complex inputs. We deem that this term will also restrict the performance in equilibrium models. We note that the symmetric constraints won't influence the final performance much as many works [32, 21] show.

A natural consideration arises as to whether we can utilize alternative kernel functions to extract input features for the equilibrium state. However, we find that other equilibrium models employing different kernels with inner products, like the polynomial kernel and sigmoid kernel, lead to a similar structure to OptEqs with appropriate weight re-parameterization and lead to similar empirical results. We provide a detailed discussion of the related models in Appendix 4.6. Thereby, we decide to use the Gaussian kernels, and our new hidden optimization equation is formulated as follows:

$$\min_{\mathbf{z}} G(\mathbf{z}; \mathbf{x}) = \min_{\mathbf{z}} \left[ \mathbf{1}^\top f(\mathbf{z}) + \frac{1}{2}\|\mathbf{z}\|_2^2 - \frac{1}{2\gamma}e^{-\gamma\|\mathbf{U}\mathbf{x}+\mathbf{b}-\mathbf{W}\mathbf{z}\|_2^2} \right], \tag{5}$$

where $\gamma$ is the hyperparameter denoting the reciprocal of Gaussian kernels' variance for scaling. Calculating $\nabla G = \mathbf{0}$ for new $G$, we can get the Gaussian kernel inspired Equilibrium models (GEQ) as the following fixed-point equation:

$$\mathbf{z}^* = \sigma\left[ e^{-\gamma\|\mathbf{U}\mathbf{x}+\mathbf{b}-\mathbf{W}\mathbf{z}^*\|_2^2} \mathbf{W}^\top(-\mathbf{W}\mathbf{z}^* + \mathbf{U}\mathbf{x} + \mathbf{b}) \right]. \tag{6}$$

Compared with linear kernels, Gaussian kernels can easily extract the non-linear relations from the input features and show more stable and powerful performance in SVM and other machine learning methods [21, 44]. We also find that the formulation of our GEQ is similar to adding a new attention module to the original equilibrium models. Therefore, our GEQ is supposed to enjoy more representative abilities than the original OptEqs. In the following parts of this section, we will analyze the theoretical advantages of our GEQ against the vanilla OptEqs. And we also empirically evaluate GEQ's performance in the following sections.

### 3.2 GEQ equals to the OptEqs with infinite width

Like other Gaussian-related models, our GEQ model can also be regarded as computing similarities by mapping them to an infinite-dimensional space. This allows GEQ to extract input features at the infinite-dimensional level, enabling the capture of non-linear dependencies in the input space. Essentially, our GEQ can be seen as a specialized version of OptEqs operating within the infinite-dimensional space after mapping input features $\mathbf{x}$ and output embedding $\mathbf{z}$ to this expanded domain.

**Proposition 1.** *The output of our GEQ (Eqn (6)) is the same as a special OptEqs' output whose hidden optimization problem is defined as follows:*

$$\min_{\mathbf{z}} G(\mathbf{z}; \mathbf{x}) = \min_{\mathbf{z}} \left[ \mathbf{1}^\top f(\mathbf{z}) + \frac{1}{2}\|\mathbf{z}\|_2^2 - \lambda \left\langle \Phi_{\mathbf{U}}(\mathbf{x} + \mathbf{U}^{-1}\mathbf{b}), \Phi_{\mathbf{W}}(\mathbf{z}) \right\rangle \right], \tag{7}$$

*where $f$ is the positive indicator function and $(1 + \partial f)^{-1}$ is the ReLU activation function, $\lambda = e^{-\gamma\|\mathbf{U}\mathbf{x}+\mathbf{b}\|_2^2}e^{-\gamma\|\mathbf{W}\mathbf{z}\|_2}$, and $\Phi_{\mathbf{W}}(\mathbf{z}) = [\mathbf{1}, \sqrt{2\gamma}\Phi_{\mathbf{W}}^{(1)}(\mathbf{z}), ..., \sqrt{(2\gamma)^i/i!}\Phi_{\mathbf{W}}^{(i)}(\mathbf{z}), ...] \in \mathbb{R}^{1\times\infty}$ which maps the inputs to the infinite-dimensional space with $\Phi_{\mathbf{W}}^{(i)} : \mathbb{R}^n \to \mathbb{R}^{in^i}$ defined as follows:*

$$\Phi_{\mathbf{W}}^{(i)} = \left[ \underbrace{\overbrace{(Wx)_0(Wx)_0...(Wx)_0}^{i}, \overbrace{(Wx)_0(Wx)_0...(Wx)_1}^{i}, ..., \overbrace{(Wx)_j(Wx)_k...(Wx)_m}^{i}, ...}_{in^i} \right], \tag{8}$$

*where $(Wx)_j$ denotes the $j$-th element of vector $\mathbf{W}\mathbf{x}$.*

Based on the analysis provided above, it becomes evident that the hidden optimization problem of our GEQ exhibits a similar formulation to a specific OptEqs, whose inputs $\mathbf{x}$ and outputs $\mathbf{z}$ are mapped to an infinite-dimensional space using the weight-tied infinite wide mapping $\Phi_{\mathbf{W}}$ and $\Phi_{\mathbf{U}}$. Given that both GEQ and OptEqs are derived from their respective hidden optimization problems, the equivalence in these problems implies the existence of the same equilibrium states for both models. Consequently, our GEQ can be considered an extension of the "infinite-depth" OptEqs to the "infinite-width" domain. Since wider neural networks are generally expected to perform better on classification tasks, we can infer that our GEQ outperforms vanilla equilibrium models like OptEqs. We further support this claim with the theoretical analysis illustrated in the subsequent sections.

### 3.3 Generalization abilities for our GEQ

Apart from the above empirical intuition, we are going to prove our GEQ's generalization advantages over OptEqs using the generalization bound under the PAC-Bayesian framework [36]. For convenience, we use $f_{geq}(\mathbf{x})$ denotes the equilibrium state $\mathbf{z}^*$ for input $\mathbf{x}$. Then we use the expected margin loss $\mathcal{L}_\eta(f_{geq}^c)$ at margin $\eta$ of our GEQ on the data distribution $\mathcal{D}$ for classification, which is defined as follows,

$$\mathcal{L}_\eta(f_{geq}^c) = \mathbb{P}_{(\mathbf{x},\mathbf{y})\sim\mathcal{D}}\left[f_{geq}^c(\mathbf{x})_y \leq \eta + \max_{j\neq y} f_{geq}^c(\mathbf{x})_j\right], \tag{9}$$

where $f_{geq}^c(\mathbf{x}) = \mathbf{W}_c f_{geq}(\mathbf{x}) + \mathbf{b}_c$ stands for GEQ's final prediction at input $\mathbf{x}$ with learnable parameters $\mathbf{W}_c$ and $\mathbf{b}_c$, and the index $j, y$ here denote the prediction score for certain class. Then we can analyze the generalization bound for our GEQ following the former work's settings [38].

**Proposition 2.** *If input $\|\mathbf{x}\|_2$ is bounded by $B$, $\mu := \max\{\|\mathbf{U}\|_2, \|\mathbf{W}\|_2, \|\mathbf{W}_c\|_2, \|\mathbf{b}\|_2\} < 1$, then we have following results for GEQ and OptEqs with ReLU activation. For $\forall\delta, \eta > 0$, with probability at least $1 - \delta$ over the training set of size $M$, we have:*

$$\mathcal{L}_0(f_{geq}^c) \leq \hat{\mathcal{L}}_\eta(f_{geq}^c) + \sqrt{\frac{16h\ln(24h)\left[\beta_{\max}\mu^4 B + (2\mu\beta_{\max} + 1)(1 - \beta_{\max}m)\mu B + (1 - \beta_{max}m)^2\right]^2 \mathcal{B}_W}{\eta^2(1 - \beta_{\max}m)^4 M}} + \frac{\ln(\frac{M\sqrt{M}}{\delta})}{M},$$
$$\mathcal{L}_0(f_{opteq}^c) \leq \hat{\mathcal{L}}_\eta(f_{opteq}^c) + \sqrt{\frac{16h\ln(24h)\left[\mu^3 B + (1 - m)\mu B + (1 - m)^2\right]^2 \mathcal{B}_W}{\eta^2(1 - m)^4 M}} + \frac{\ln(\frac{M\sqrt{M}}{\delta})}{M}, \tag{10}$$

*where $\hat{\mathcal{L}}_\eta(f_{geq}^c)$ denotes the empirical margin loss on the training set, the maximum scaling number is defined by $\beta_{max} := \max_{\mathbf{x}\in\mathcal{D}} e^{-\gamma\|\mathbf{U}\mathbf{x}+\mathbf{b}-\mathbf{W}\mathbf{z}\|_2^2}$, $\mathcal{B}_W := \|\mathbf{W}^\top\mathbf{W}\|_F^2 + \|\mathbf{U}\|_F^2 + \|\mathbf{b}\|_2^2 + \|\mathbf{W}_c\|_F^2 + \|\mathbf{b}_c\|_2^2$, and $m = \|\mathbf{W}^\top\mathbf{W}\|_2$ is less than 1 to ensure the convergence of equilibrium models.*

**Remark 1.** *If $\beta_{\max} < 0.8$ and $\mu, m > 0.9$, we can get $\frac{\beta_{\max}\mu}{1-\beta_{\max}m} < \frac{1}{1-m}$ and $\frac{2\mu\beta_{\max}+1}{1-\beta_{\max}m} \leq \frac{1}{1-m}$. In the meanwhile, our GEQ's generalization bound is tighter than the original OptEq.*

In practical experiments, we find that the above conditions for $\beta_{\max}$ and $\mu, m$ are satisfied in most cases. And the following experiments also support our above theoretical advantages. However, the above bound is not tight, and how to approximate a much tighter bound for equilibrium models still needs exploring.

### 3.4 GEQ enjoys More Stable Performance

Apart from better performance, Gaussian kernel stands out as one of the most extensively employed kernels in machine learning tasks owing to its stability across various input scenarios. Motivated by this, we aim to investigate whether incorporating Gaussian kernels into our equilibrium models can enhance the model's stability across diverse inputs. Firstly, we are going to estimate output changes with respect to the input perturbations.

**Proposition 3.** *If norms for the inputs and outputs are bounded by $B$, the spectral norm for the weight parameter $\mathbf{W}, \mathbf{U}$ of equilibrium models with ReLU activation are bounded by $\mu < 1$ to ensure*

*convergence, then we have the conclusions as below:*

$$\|f_{geq}(\mathbf{x}_1) - f_{geq}(\mathbf{x}_2)\|_2 \leq L_{geq}\|\mathbf{x}_1 - \mathbf{x}_2\|_2 = \frac{\beta_{\max}\mu^2 + \sqrt{\gamma}B\mu^3}{1 - \beta_{\max}\mu^2 - \sqrt{\gamma}B\mu^3}\|\mathbf{x}_1 - \mathbf{x}_2\|_2, \quad (11)$$

$$\|f_{opteq}(\mathbf{x}_1) - f_{opteq}(\mathbf{x}_2)\|_2 \leq L_{opteq}\|\mathbf{x}_1 - \mathbf{x}_2\|_2 = \frac{\mu}{1 - \mu^2}\|\mathbf{x}_1 - \mathbf{x}_2\|_2, \quad (12)$$

*where $\mathbf{x}_1$ and $\mathbf{x}_2$ are input samples, $f_{geq}(\mathbf{x}.)$ and $f_{opteq}(\mathbf{x}.)$ denotes the equilibrium states for GEQ and OptEqs given input $\mathbf{x}.$, and $\beta_{max} := \max_{\mathbf{x}\in\mathcal{D}} e^{-\gamma\|\mathbf{U}\mathbf{x} - \mathbf{W}\mathbf{z}\|_2^2} < 1$.*

**Remark 2.** *If $\beta_{\max} < 0.8$, $B < 1$, and $\sqrt{\gamma} < 0.2$, then $L_{geq} < L_{opteq}$.*

In practical experiments, we choose different $\gamma$ to reach the above condition for $\beta_{\max}$ and the condition for input $B$ can also be achieved by normalization layers. Although the above bound is not tight, we can use it as a rough explanation for our GEQ's stability, which is demonstrated in the following experiments. How to approximate a much tighter bound for equilibrium models still needs exploring.

Besides having stable outputs under perturbations, a stable model should also show large output differences for different classes to make classification easier. However, the above Lipschitz term can not constrain outputs' similarity when samples are far apart, then we need a new metric for analysis. In line with previous works [18, 34, 10], we assume all weight parameters go to infinite dimensions and analyze the expected output similarity $\kappa$ for a model $f$ for inputs $\mathbf{x}_1$ and $\mathbf{x}_2$ defined below:

$$\kappa(\mathbf{x}_1, \mathbf{x}_2) = \mathbb{E}\left[f(\mathbf{x}_1)^\top f(\mathbf{x}_2)\right] = \int_{\mathbb{R}} f_{\mathbf{u}}(\mathbf{x}_1)^\top f_{\mathbf{u}}(\mathbf{x}_2)p(\mathbf{u})d\mathbf{u}, \quad (13)$$

with $p(\mathbf{u})$ is the distribution of weight $\mathbf{U}$'s vectorization. If $\kappa$ is smaller for samples $\mathbf{x}_1$ and $\mathbf{x}_2$ when they belong to different classes, which means they are far away, then the classifier can easily classify these two samples with different labels. The margin for the classification will also be large and easy for the classification of difficult samples. The $\kappa$'s upper bound for GEQ and OptEqs are listed below:

**Proposition 4.** *If norms for the inputs and outputs are bounded by $B$, the spectral norm for the weight parameter $\mathbf{W}$ of equilibrium models with ReLU activation are bounded by $\mu < 1$ to ensure the convergence, and each row in $\mathbf{U}$ obeys the spherical Gaussian distributions $\mathcal{N}(0, \mathbb{E}[U_i^2]\mathbf{I})$. Then we have the following conclusions for the expectation of the output similarity for GEQ and OptEqs with respect to input $\mathbf{x}_1, \mathbf{x}_2$ as follows,*

$$\kappa_{geq}(\mathbf{x}_1, \mathbf{x}_2) \leq \overline{\kappa}_{geq} = \frac{\mu^2 D e^{-\frac{\gamma}{4}(\sigma_{\min}(\mathbf{U})^2\|\mathbf{x}_1-\mathbf{x}_2\|_2^2)}\mathbb{E}[U_i^2]\|\mathbf{x}_1\|_2\|\mathbf{x}_2\|_2\left(\sin\theta_0 + (\pi - \theta_0)\cos\theta_0\right)}{2\pi(1 - \beta_{\max}\mu^2)^2},$$

$$\qquad(14)$$

$$\kappa_{opteq}(\mathbf{x}_1, \mathbf{x}_2) \leq \overline{\kappa}_{opteq} = \frac{\mathbb{E}[U_i^2]\|\mathbf{x}_1\|_2\|\mathbf{x}_2\|_2\left(\sin\theta_0 + (\pi - \theta_0)\cos\theta_0\right)}{2\pi(1 - \mu^2)^2}, \quad (15)$$

*where $\mathbf{x}_1$ and $\mathbf{x}_2$ are input samples, $D = e^{\gamma B\|\mathbf{W}\|_2^2}$, $\beta_{max} := \max_{\mathbf{x}\in\mathcal{D}} e^{-\gamma\|\mathbf{U}\mathbf{x} - \mathbf{W}\mathbf{z}\|_2^2}$, $\sigma_{\min}(\mathbf{U})$ is $\mathbf{U}$'s minimal singular term, and $\theta_0 = \cos^{-1}(\frac{\langle\mathbf{x}_1, \mathbf{x}_2\rangle}{\|\mathbf{x}_1\|\|\mathbf{x}_2\|})$ is the angle between the samples.*

**Remark 3.** *If $\|\mathbf{x}_1 - \mathbf{x}_2\|_2 \geq 2\sqrt{-log(1/D)}/\sigma_{\min}(\mathbf{U})$, then $\overline{\kappa}_{geq} \leq \overline{\kappa}_{geq}$.*

Based on the aforementioned analysis, it is evident that our GEQ exhibits a smaller output similarity for dissimilar samples. As a result, the predictions made by GEQ are primarily based on the most similar samples, enabling it to successfully classify challenging instances. This claim is further supported by the results obtained from our carefully designed experiments.

### 3.5 Patch Splitting in GEQ

Since different parts of images have different impacts on the image classification, calculating the whole similarity using Gaussian kernels for GEQ is not enough. Inspired by former works [47], we also split the feature map $\mathbf{U}\mathbf{x}$ into patches, and then our optimization problem becomes:

$$\min_{\mathbf{z}} G(\mathbf{z}; \mathbf{x}) = \min_{\mathbf{z}}\left[\mathbf{1}^\top f(\mathbf{z}) + \frac{1}{2}\|\mathbf{z}\|_2^2 - \frac{1}{2\gamma}\sum_{i=1}^N e^{-\gamma\|(\tilde{\mathbf{x}}_i - \tilde{\mathbf{z}}_i)\mathbf{W}_h\|_2^2}\right], \quad (16)$$

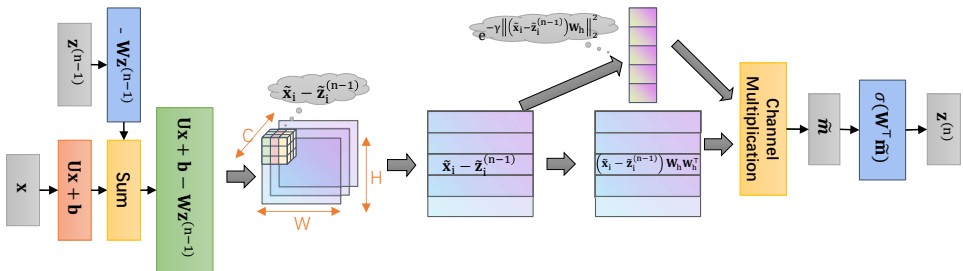

Figure 1: The sketch map of one layer GEQ's $n$-th fixed point iteration. $\mathbf{x}$ is the input and $\mathbf{z}^{(n-1)}, \mathbf{z}^{(n)}$ are the output of $(n-1)$-th, $n$-th iteration, and $\tilde{\mathbf{m}}$ is a middle state.

where $\tilde{\mathbf{x}}_i \in \mathbb{R}^{c_s p^2}$ is the $i$-th patch of $\mathbf{Ux} + \mathbf{b}$ while $\tilde{\mathbf{z}}_i \in \mathbb{R}^{c_s p^2}$ is the $i$-th patch of $\mathbf{Wz}$ and $\mathbf{W}_h \in \mathbb{R}^{c_s p^2 \times c_{hid}}$ is a linear layer to project patches with different size to the constant dimension. $c_s$ denotes the channel splitting number, $p$ denotes the patch size, and $c_{hid}$ denotes the hidden dimension of patches after projection. We note that the patch-splitting approach is a GEQ's unique feature, as incorporating this technique makes no difference in OptEqs due to its linear kernel. Figure 1 provides a sketch for GEQ's $i$-th fixed-point iteration. From the figure, it is evident that our GEQ can be viewed as a special OptEqs with additional attention mechanisms to capture the most important regions. Thereby, it can achieve enhanced performance. For a more detailed understanding of the forward procedure in our GEQ, please refer to Appendix A.1.

# 4 Empirical Results

## 4.1 Experiment Settings

In our experiments, we employed parallel GEQs with different input scales like MOptEqs and averaged the output of each branch after average pooling or nearest up-sampling to fuse the branches. We use weight normalization to ensure the convergence as MOptEqs and MDEQ, and set $\gamma$ to $0.2/M$, where $M$ is the minimum $\|\tilde{\mathbf{x}}. - \tilde{\mathbf{z}}.\mathbf{W}_h\|_2^2$ among all patches. For the equilibrium calculation, we used the Anderson algorithm in the forward procedure, similar to other implicit models [28], and applied Phantom gradients [14] for back-propagation. All models were trained using SGD with a step learning rate schedule. We implemented our experiments on the PyTorch platform [39] using RTX-3090. Further details can be found in the Appendix A.6. To compare the performance of our GEQ, we used MOptEqs and MDEQ as benchmark implicit models, which have demonstrated superior performance over OptEqs on image classification tasks. Additionally, we used ResNet-18 and ResNet-50 as benchmark explicit models for comparison.

## 4.2 Results for Image Classification

Firstly, we finish the experiments on CIFAR-10 and CIFAR-100. They are widely used datasets for image classification on small images. In the experiment, we parallel 6 branches GEQ with the input scale is $32, 16, 8, 8, 4, 4$ and MOptEqs' architecture setting is also the same. The details can be found in the Appendix. As for the comparison, we also conduct experiments of the same training procedure for MDEQ, MOptEqs, and ResNet. The results are listed in Table 1. From the results, one can see that our GEQ enjoys clear advantages on CIFAR datasets, which demonstrates the powerful generalization ability of other models.

Besides small datasets, we also conducted experiments on large-scale image datasets, as presented in Table 2. The results clearly demonstrate the consistent superiority of our GEQ over other models, highlighting its clear advantages. Particularly noteworthy is our GEQ achieves a $2\%$ improvement on ImageNet-100 against deep model ResNet-50 while consuming approximately half the number of parameters, which emphasizes the effectiveness and efficiency of GEQ on large-scale inputs.

|          | Model Size | Accuracy |
|----------|------------|----------|
| ResNet-18 | 10M | $93.5 \pm 0.2\%$ |
| ResNet-50 | 23M | $95.2 \pm 0.2$ |
| MDEQ | 10M | $94.2 \pm 0.3\%$ |
| MOptEqs | 8M | $94.6 \pm 0.2\%$ |
| GEQ | 5M | $94.8 \pm 0.1\%$ |
| GEQ | 8M | $\mathbf{95.6 \pm 0.2}\%$ |

(a) CIFAR-10.

|          | Model Size | Accuracy |
|----------|------------|----------|
| ResNet-18 | 10M | $74.5 \pm 0.2\%$ |
| ResNet-50 | 23M | $77.9 \pm 0.1\%$ |
| MDEQ | 10M | $74.7 \pm 0.3\%$ |
| MOptEqs | 8M | $75.6 \pm 0.2\%$ |
| GEQ | 5M | $76.4 \pm 0.3\%$ |
| GEQ | 8M | $\mathbf{78.2 \pm 0.2}\%$ |

(b) CIFAR-100.

Table 1: The Empirical results for image classification on CIFAR-10 and CIFAR-100.

|          | Model Size | Accuracy |
|----------|------------|----------|
| ResNet-18 | 11M | $92.3 \pm 0.1\%$ |
| ResNet-50 | 23M | $93.0 \pm 0.2\%$ |
| MDEQ | 10M | $91.5 \pm 0.2\%$ |
| MOptEqs | 10M | $92.4 \pm 0.2\%$ |
| GEQ | 6M | $\mathbf{92.9 \pm 0.2}\%$ |
| GEQ | 13M | $\mathbf{93.2 \pm 0.1}\%$ |

(a) ImageNette.

|          | Model Size | Accuracy |
|----------|------------|----------|
| ResNet-18 | 11M | $80.9 \pm 0.3\%$ |
| ResNet-50 | 23M | $81.7 \pm 0.2\%$ |
| MDEQ | 10M | $81.3 \pm 0,2\%$ |
| MOptEqs | 13M | $81.5 \pm 0.4\%$ |
| GEQ | 6M | $\mathbf{82.2 \pm 0.2}\%$ |
| GEQ | 13M | $\mathbf{83.9 \pm 0.3}\%$ |

(b) ImageNet-100.

Table 2: The Empirical results for image classification on ImageNette and ImageNet-100.

## 4.3 Validations on the models' stability

**Evaluation on Unseen difficult samples.** In order to assess the stability of our GEQ model on difficult examples, we conducted experiments using CIFAR-100 super-class classification. CIFAR-100 consists of 20 super classes, each containing five sub-classes [2]. We trained our GEQ and MOptEqs models to predict the super-classes using the first four sub-classes from each super-class for training. We evaluated the models using both the test set, which includes the first four sub-classes from each super-class (referred to as "Known Accuracy"), and a separate set of samples from unseen sub-classes (referred to as "Unknown Accuracy"). The classification of the unseen samples is more difficult as they are different from the training set. The results of our GEQ and MOptEqs models are presented in Table 3.

|          | Known Accuracy | Unknown Accuracy |
|----------|----------------|------------------|
| MOptEqs | $80.1 \pm 0.3\%$ | $77.4 \pm 0.5\%$ |
| GEQ | $\mathbf{80.9 \pm 0.2}\%$ | $\mathbf{80.1 \pm 0.6}\%$ |

Table 3: Empirical rsults on CIFAR-100's super-class classification.

The above table clearly demonstrates that our GEQ model surpasses MOptEqs in achieving superior performance on the challenging task at hand and demonstrates GEQ's stability. Such advantages can be attributed to the fact that GEQ exhibits smaller output similarities compared to OptEqs when input samples are far apart (e.g., samples from different classes). This characteristic can lead to larger margins between different classes, enabling the classifier to be more easily optimized during training. Consequently, our GEQ model excels in accurately classifying difficult unseen samples, further highlighting its stability and superiority over former equilibrium models.

**Ablation Studies on corrupted datasets.** Apart from difficult samples, we are going to compare the robustness of our GEQ, MOptEqs, and ResNet on the CIFAR-10 corruption dataset, which contains 19 common corruptions including image transformation, image blurring, weather, and digital noises on CIFAR's test datasets. Average results on 5 degrees CIFAR-10 corruption datasets list in Figure 2.

From the result, one can see that our GEQ based on Gaussian kernels is more robust than MOptEqs and ResNet. In particular, our GEQ can show better performance against structured noise and some image transformation. The above results also demonstrate the stability of our GEQ structure.

---

[2]For example, super class "people" contains five sub classes:"baby","girl","man","man","woman"

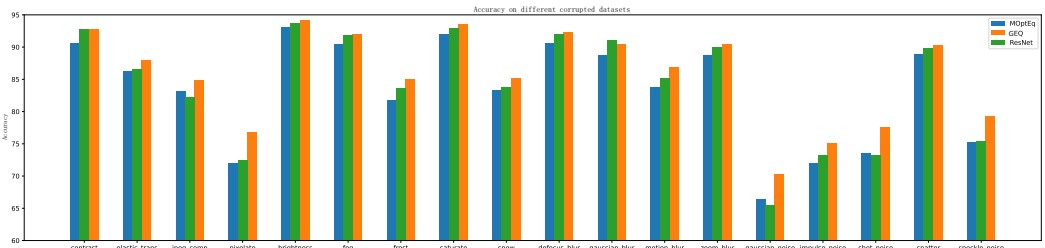

Figure 2: The results for different models under different corruptions.

## 4.4 Ablation Studies on Saliency Map

The saliency maps generated by GradCAM [45] offer valuable insights into the visual attention of both MOptEqs and GEQ models. These maps highlight the regions of the image that are crucial for the model's predictions. Figure 3 presents the saliency maps obtained for an image from the ImageNet dataset using both models. Upon observation, it becomes evident that GEQ exhibits a higher degree of focus on the significant regions directly associated with the predicted label "manta". In contrast, MOptEqs tends to allocate attention to unrelated regions such as the shells. This discrepancy indicates that the attention-like module induced by the Gaussian kernel in GEQ enhances the concentration of the model's attention, resulting in improved performance compared to MOptEqs.

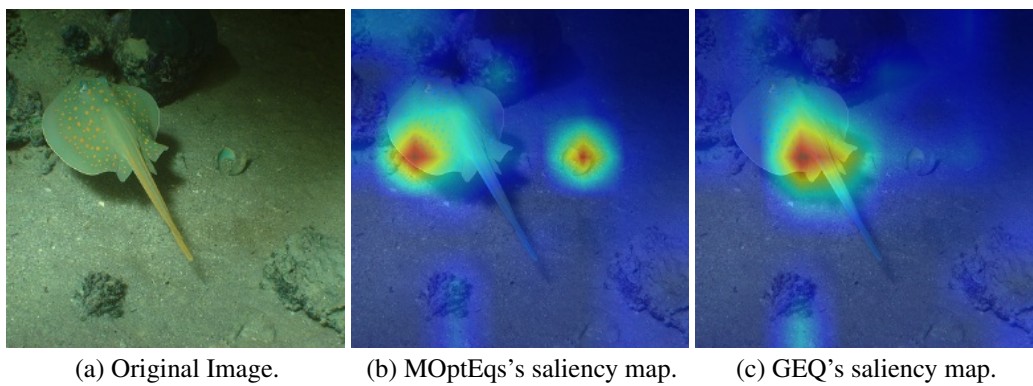

(a) Original Image.        (b) MOptEqs's saliency map.        (c) GEQ's saliency map.

Figure 3: Saliency map for GEQ and MOptEqs on the ImageNet image.

## 4.5 Ablation Studies on Patch splitting

The performance of our GEQ model is influenced by the channel splitting parameter and the patch size. Choosing large values for these parameters causes the kernel to focus mainly on global information while selecting small values makes the kernel concentrate on local features. To understand the impact of these choices on model performance, we conducted experiments, the results of which are presented in Figure 4. This figure illustrates the relationship between the channel splitting parameter, patch size, and the model's performance. By analyzing these results, we gain insights into the optimal values for these parameters that yield the best performance for our GEQ model.

The accuracy trend depicted in the figure shows an initial increase followed by a decrease as the channel split and patch size increased. Based on these empirical results, we select a patch size of 2 and a channel split of 8 for both the CIFAR and ImageNet experiments. These parameter choices are made to optimize the performance of our models on the respective datasets.

## 4.6 Comparison with other kernel functions

Firstly, we introduce different commonly used kernels, such as polynomial, sigmoid, and Gaussian kernels, to reformulate the hidden optimization problem for equilibrium models. Table 4 illustrates the equilibrium models induced by these kernels. It can be observed that equilibrium models with polynomial and sigmoid kernels also incorporate new attentive modules. However, their attentive

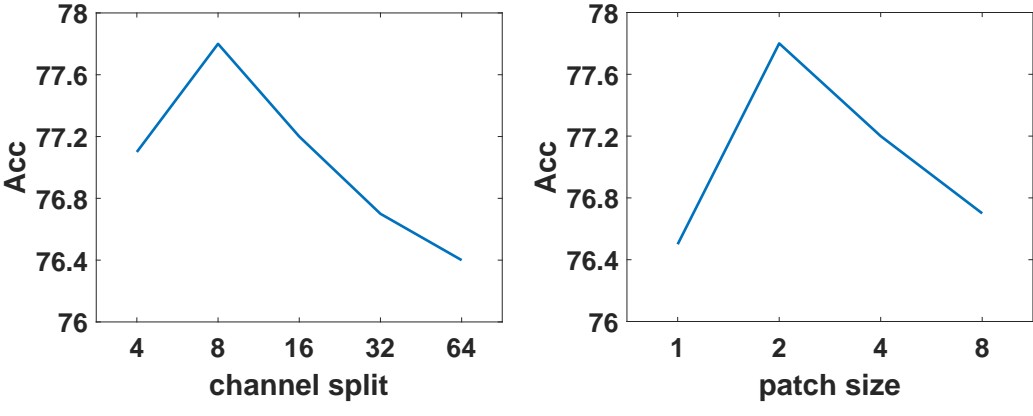

(a) Influence of the channel splitting parameter.  (b) Influence of the patch size parameter.

Figure 4: The influence on the patch size and the channel splitting parameter for our GEQ on CIFAR-100 datasets.

kernels only constrain the input features $\mathbf{Ux} + \mathbf{b}$ and do not directly affect the activation of the output $\mathbf{z}^*$. As a result, their performance may be inferior to our GEQ model. To validate these claims, we evaluate the performance of different models on the CIFAR-100 dataset. Since the fusion module is the primary difference between MOptEqs and OptEqs, we can easily modify the structure of MOptEqs to include different kernel-induced attentive modules using the equations in Table 4, resulting in MOptEqs (Polynomial) and MOptEqs (Sigmoid). The results are presented in Table 5, which clearly demonstrates the superior performance of our GEQ model. For a more in-depth analysis of GEQ, we refer readers to the main paper.

| Kernel | Hidden Optimization Problem | Equilibrium Model |
|---|---|---|
| Linear | $\min_{\mathbf{z}}\left[\mathbf{1}^\top f(\mathbf{z}) + \frac{1}{2}\|\mathbf{z}\|_2^2 - \langle \mathbf{Ux}+\mathbf{b}, \mathbf{z}\rangle - \frac{1}{2}\|\mathbf{Wz}\|_2^2\right]$ | $\mathbf{z}^* = \sigma\left(\mathbf{W}^\top \mathbf{Wz}^* + \mathbf{Ux} + \mathbf{b}\right)$ |
| Polynomial | $\min_{\mathbf{z}}\left[\mathbf{1}^\top f(\mathbf{z}) + \frac{1}{2}\|\mathbf{z}\|_2^2 - (\langle \mathbf{Ux}+\mathbf{b}, \mathbf{z}\rangle)^d - \frac{1}{2}\|\mathbf{Wz}\|_2^2\right]$ | $\mathbf{z}^* = \sigma\left(\mathbf{W}^\top \mathbf{Wz}^* + d\left(\langle \mathbf{Ux}+\mathbf{b}, \mathbf{z}\rangle\right)^{d-1}\left(\mathbf{Ux}+\mathbf{b}\right)\right)$ |
| Sigmoid | $\min_{\mathbf{z}}\left[\mathbf{1}^\top f(\mathbf{z}) + \frac{1}{2}\|\mathbf{z}\|_2^2 - \tanh\left(\langle \mathbf{Ux}+\mathbf{b}, \mathbf{z}\rangle\right) - \frac{1}{2}\|\mathbf{Wz}\|_2^2\right]$ | $\mathbf{z}^* = \sigma\left(\mathbf{W}^\top \mathbf{Wz}^* + \left(1 - \tanh^2\left(\langle \mathbf{Ux}+\mathbf{b}, \mathbf{z}\rangle\right)\right)\left(\mathbf{Ux}+\mathbf{b}\right)\right)$ |
| Gaussian | $\min_{\mathbf{z}}\left[\mathbf{1}^\top f(\mathbf{z}) + \frac{1}{2}\|\mathbf{z}\|_2^2 - \frac{1}{2\gamma}e^{-\gamma\|\mathbf{Ux}+\mathbf{b}-\mathbf{Wz}\|_2^2}\right]$ | $\mathbf{z}^* = \sigma\left[e^{-\gamma\|\mathbf{Ux}+\mathbf{b}-\mathbf{Wz}^*\|_2^2}\mathbf{W}^\top\left(-\mathbf{Wz}^* + \mathbf{Ux} + \mathbf{b}\right)\right]$ |

Table 4: The hidden optimization problems and their related equilibrium models. $d > 1$ is an integer denoting the polynomial order.

|  | Model Size | Accuracy |
|---|---|---|
| MOptEqs | 8M | $75.6 \pm 0.2\%$ |
| MOptEqs (Polynomial) | 8M | $75.1 \pm 0.4\%$ |
| MOptEqs (Sigmoid) | 8M | $76.1 \pm 0.3\%$ |
| GEQ | 8M | $\mathbf{78.2 \pm 0.2\%}$ |

Table 5: Comparison of equilibrium models with different kernel functions on CIFAR-100.

## 5   Conclusions

In this paper, we introduce a novel optimization-induced equilibrium model called GEQ, which utilizes Gaussian kernels in its optimization-induced framework. Our model incorporates a new attentive module that arises from its novel hidden optimization problem formulation. Notably, GEQ exhibits significantly improved performance in classification tasks, outperforming deep models as well. Moreover, GEQ can be interpreted as a weight-tied model with infinite width and depth, highlighting its expressive power. We also provide theoretical analysis demonstrating the superiority of our models in terms of generalization ability and stability compared to previous OptEqs. Empirical results further validate the effectiveness of our proposed approach.

## Acknowledgments

Zhouchen Lin was supported by National Key R&D Program of China (2022ZD0160302), the major key project of PCL, China (No. PCL2021A12), the NSF China (No. 62276004), and Qualcomm. Yisen Wang was supported by National Natural Science Foundation of China (62006153, 62376010, 92370129), Open Research Projects of Zhejiang Lab (No. 2022RC0AB05), and Beijing Nova Program (20230484344).

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

# A Appendix

## A.1 Forward Procedure for GEQ

The pseudo-code for our GEQ is listed in Algorithm 1.

---

**Algorithm 1:** Calculating one layer GEQ.

---

**Require:** initial state $\mathbf{z}^{(0)}$, weight parameter $\mathbf{W}$, $\mathbf{Ux} \in \mathbb{R}^{chw}$, channel split $c_s$, patch size $p$, hidden layer
  $\mathbf{W}_h \in \mathbb{R}^{c_s p^2 \times 32}$

**Ensure:** Get the output $\mathbf{z}^*$ of $i$-th fixed point iteration.

    def $g(\mathbf{z}^{(i)}; \mathbf{x}, \mathbf{U}, \mathbf{W}, \mathbf{b})$:

        Rearrage $\mathbf{Wz}^{(i)} \to \tilde{\mathbf{z}}$, $\mathbf{Ux} + \mathbf{b} \to \tilde{\mathbf{x}} \in \mathbb{R}^{\frac{chw}{c_s p^2} \times (c_s p^2)}$

        $\tilde{\mathbf{m}} = diag\left(e^{-\gamma \|(\tilde{\mathbf{x}}. - \tilde{\mathbf{z}}.)\mathbf{W}_h\|_2^2}\right) (\tilde{\mathbf{x}} - \tilde{\mathbf{z}}) \mathbf{W}_h \mathbf{W}_h^\top$

        Rearrage $\tilde{\mathbf{m}} \to \mathbf{m} \in \mathbb{R}^{c \times h \times w}$

        return $\mathbf{z}^{(i+1)} = \sigma\left(\mathbf{W}^\top \mathbf{m}\right)$

    with torch.no_grad():

        Use anderson algorithm to solve $\mathbf{z}^* = g(\mathbf{z}^*; \mathbf{x}, \mathbf{U}, \mathbf{W}, \mathbf{b})$

    # calculate gradient via phantom gradient:

    **for** i in range(5) **do**

      $\mathbf{z}^* = 0.2 \times \mathbf{z}^* + 0.8 \times g(\mathbf{z}^*; \mathbf{x}, \mathbf{U}, \mathbf{W}, \mathbf{b})$.

    **end for**

    **return $\mathbf{z}^*$**

---

## A.2 Proofs for proposition 1

**Proposition 1.** *The outputs of GEQ with Gaussian kernels (Eqn (6)) is the same as Optimized induced Equilibrium Models' output whose output is the optimal solution of the hidden optimization problems:*

$$\min_{\mathbf{z}} G(\mathbf{z}; \mathbf{x}) = \mathbf{1}^\top f(\mathbf{z}) + \frac{1}{2}\|\mathbf{z}\|^2 - \lambda \left\langle \Phi_{\mathbf{U}}(\mathbf{x} + \mathbf{U}^{-1}\mathbf{b}), \Phi_{\mathbf{W}}(\mathbf{z}) \right\rangle \tag{17}$$

*where $\Phi_{\mathbf{W}}(\mathbf{z}) = \left[\mathbf{1}, \sqrt{2\gamma}\Phi_{\mathbf{W}}^{(1)}(\mathbf{z}), ..., \sqrt{\frac{(2\gamma)^i}{i}}\Phi_{\mathbf{W}}^{(i)}(\mathbf{z}), ...\right] \in \mathbb{R}^{1 \times \infty}$ which projects the features to the infinite-dimensional space. And $\Phi_{\mathbf{W}}^{(i)} : \mathbb{R}^n \to \mathbb{R}^{in^i}$ is the $k$-tuple permutation with repetitions formulated as follows:*

$$\Phi_{\mathbf{W}}^{(i)} = \left[\underbrace{\overbrace{(Wx)_1(Wx)_1...(Wx)_1}^{i}, \overbrace{(Wx)_1(Wx)_1...(Wx)_2}^{i}, ..., \overbrace{(Wx)_j(Wx)_k...(Wx)_m}^{i}, ...}_{in^i}\right] \tag{18}$$

*where $(Wx)_j$ denotes the $j$-th element of vector $\mathbf{Wx}$. Then the Gaussian kernel can also be regarded as calculating the input features after the weight-tied infinite wide projection $\Phi_{\mathbf{W}}$ and $\Phi_{\mathbf{U}}$.*

*Proof.* We can formulate the OptEqs' hidden optimization problem with Gaussian kernels as below:

$$\min_{\mathbf{z}} G(\mathbf{z}; \mathbf{x}) = \mathbf{1}^\top f(\mathbf{z}) + \frac{1}{2}\|\mathbf{z}\|^2 - \frac{1}{2\gamma}e^{-\gamma\|\mathbf{Ux}+\mathbf{b}-\mathbf{Wz}\|_2^2}, \tag{19}$$

For $e^{-\gamma\|(\mathbf{Ux}+\mathbf{b}-\mathbf{Wz})\|_2^2}$, we have

$$\begin{aligned}
e^{-\gamma\|\mathbf{Ux}+\mathbf{b}-\mathbf{Wz}\|_2^2} &= e^{-\gamma\|\mathbf{Ux}+\mathbf{b}\|_2^2 - \gamma\|\mathbf{Wz}\|_2^2 + 2\gamma\langle\mathbf{Ux}+\mathbf{b}, \mathbf{Wz}\rangle}, \\
&= e^{-\gamma\|\mathbf{Ux}+\mathbf{b}\|_2^2 - \gamma\|\mathbf{Wz}\|_2} e^{2\gamma\langle\mathbf{Ux}+\mathbf{b}, \mathbf{Wz}\rangle},
\end{aligned} \tag{20}$$

letting $\lambda = e^{-\gamma\|\mathbf{Ux}+\mathbf{b}\|_2^2} e^{-\gamma\|\mathbf{Wz}\|_2}$ and we do the Taylor expansion for $e^{\langle\mathbf{Ux}+\mathbf{b}, \mathbf{Wz}\rangle}$, we have:

$$e^{-\gamma\|\mathbf{Ux}+\mathbf{b}-\mathbf{Wz}\|_2^2} = \lambda \sum_{i=0}^{\infty} \frac{\left(\langle\sqrt{2\gamma}(\mathbf{Ux}+\mathbf{b}), \sqrt{2\gamma}(\mathbf{Wz})\rangle\right)^i}{i!}. \tag{21}$$

For any $i$ we have from the permutation theory:

$$\frac{\left(\langle\sqrt{2\gamma}(\mathbf{U}\mathbf{x}+\mathbf{b}),\sqrt{2\gamma}(\mathbf{W}\mathbf{z})\rangle\right)^i}{i!} = \lambda\left\langle\Phi_{\mathbf{U}}^{(i)}(\mathbf{x}+\mathbf{U}^{-1}\mathbf{b}),\Phi_{\mathbf{W}}^{(i)}(\mathbf{z})\right\rangle, \tag{22}$$

where $\Phi_{\mathbf{W}}^{(i)}(\mathbf{x})$ is the $i$-tuple permutation with the repetition for given $(Wx)_1,(Wx)_2,...,(Wx)_n$. Each element of $\Phi_{\mathbf{W}}^{(i)}(\mathbf{x})$ is one possible permutation. Since there are $n^i$ tuples, then $\Phi_{\mathbf{W}}^{(i)}$ can project the input features to $in^i$ space as follows,

$$\Phi_{\mathbf{W}}^{(i)} = \left[\underbrace{\overbrace{(Wx)_1(Wx)_1...(Wx)_1}^{i},\overbrace{(Wx)_1(Wx)_1...(Wx)_2}^{i},...,\overbrace{(Wx)_j(Wx)_k...(Wx)_m}^{i},...}_{in^i}\right]. \tag{23}$$

Thereby, the hidden optimization problem for our GEQ can be reformulated as,

$$\min_{\mathbf{z}} G(\mathbf{z};\mathbf{x}) = \mathbf{1}^\top f(\mathbf{z}) + \frac{1}{2}\|\mathbf{z}\|^2 - \lambda\left\langle\Phi_{\mathbf{U}}(\mathbf{x}+\mathbf{U}^{-1}\mathbf{b}),\Phi_{\mathbf{W}}(\mathbf{z})\right\rangle \tag{24}$$

Then our conclusion is proved. $\qquad\square$

### A.3 Proofs for proposition 2

**Proposition 2.** *If input $\|\mathbf{x}\|_2$ is bounded by $B$, $\mu := \max\{\|\mathbf{U}\|_2,\|\mathbf{W}\|_2,\|\mathbf{W}_c\|_2,\|\mathbf{b}\|_2\} < 1$, then we have following results for GEQ and OptEqs with ReLU activations. For $\forall\delta,\eta > 0$, with probability at least $1-\delta$ over the training set of size M, we have:*

$$\mathcal{L}_0(f_{geq}^c) \leq \hat{\mathcal{L}}_\eta(f_{geq}^c) + \sqrt{\frac{16h\ln(24h)\left[\beta_{\max}\mu^4 B + (2\mu\beta_{\max}+1)(1-\beta_{\max}m)\mu B + (1-\beta_{max}m)^2\right]^2\mathcal{B}_W}{\eta^2(1-\beta_{\max}m)^4 M}} + \frac{\ln(\frac{M\sqrt{M}}{\delta})}{M},$$

$$\mathcal{L}_0(f_{opteq}^c) \leq \hat{\mathcal{L}}_\eta(f_{opteq}^c) + \sqrt{\frac{16h\ln(24h)\left[\mu^3 B + (1-m)\mu B + (1-m)^2\right]^2\mathcal{B}_W}{\eta^2(1-m)^4 M}} + \frac{\ln(\frac{M\sqrt{M}}{\delta})}{M}, \tag{25}$$

*where $\hat{\mathcal{L}}_\eta(f_{geq}^c)$ denotes the empirical margin loss on the training set, the maximum scaling number is defined by $\beta_{max} := \max_{\mathbf{x}\in\mathcal{D}} e^{-\gamma\|\mathbf{U}\mathbf{x}+\mathbf{b}-\mathbf{W}\mathbf{z}\|_2^2}$, $\mathcal{B}_W := \|\mathbf{W}^\top\mathbf{W}\|_F^2 + \|\mathbf{U}\|_F^2 + \|\mathbf{b}\|_2^2 + \|\mathbf{W}_c\|_F^2 + \|\mathbf{b}_c\|_2^2$, and $m = \|\mathbf{W}^\top\mathbf{W}\|_2$ is less than 1 to ensure the convergence of equilibrium models.*

Before the proof our results, we need to introduce a lemma in former work [38] for the perturbation bound for GEQs and reformulated OptEqs as follows.

**Lemma 1.** *Let $\|\mathbf{W}\|_2 \leq m$ and $\|\overline{\mathbf{W}}\|_2 \leq m$. Then change in the output of the DEqs $\mathbf{z} = \sigma(\mathbf{W}\mathbf{z}+\mathbf{U}\mathbf{x}+\mathbf{b})$ on perturbation the weights and biases from $\mathbf{W},\mathbf{U},\mathbf{b}$ to $\overline{\mathbf{W}},\overline{\mathbf{U}},\overline{\mathbf{b}}$ is bounded as follows:*

$$\left\|f(\overline{\mathbf{W}}\mathbf{z}+\overline{\mathbf{U}}\mathbf{x}+\overline{\mathbf{b}}) - f(\mathbf{W}\mathbf{z}+\mathbf{U}\mathbf{x}+\mathbf{b})\right\|_2 \leq$$
$$\frac{\left\|\overline{\mathbf{W}}-\mathbf{W}\right\|_2\|\mathbf{U}\mathbf{x}+\mathbf{b}\|_2 + \left\|(\overline{\mathbf{U}}-\mathbf{U})\mathbf{x}\right\|_2 + \|\overline{\mathbf{b}}-\mathbf{b}\|_2}{(1-m)^2} \tag{26}$$

Like former works [38], we we also introduce another lemma [36] here:

**Lemma 2.** *Let $f_w$ be any predictor with parameters $w$, and let $P$ denote any distribution on the parameters that are independent of the training data. Then, for any $\delta,\gamma > 0$, with probability $\geq 1-\delta$ over the training data of size M, for any $w$, and any random perturbation $u$ such that $\mathbb{P}\left[\max_x\|f_{w+u}(x)-f_w(x)\|_\infty < \frac{\eta}{4}\right] \geq \frac{1}{2}$, we have*

$$\mathcal{L}_0(f_w) \leq \hat{\mathcal{L}}_\eta f_w + 4\sqrt{\frac{KL(w+u\|P)+\ln\frac{6M}{\delta}}{M-1}} \tag{27}$$

Then we can derive the perturbation bound for the reformulated OptEqs and our GEQ following former settings [38]. First, we also incorporate a fully connected layer at the end as we mentioned in the paper.

$$f_{geq}^c(\mathbf{x}) = \mathbf{W}_c f_{geq}(\mathbf{x}) + \mathbf{b}_c, f_{opteq}^c(\mathbf{x}) = \mathbf{W}_c f_{opteq}(\mathbf{x}) + \mathbf{b}_c. \tag{28}$$

Since the entries in the perturbations obeying the distribution os $\mathcal{N}(0, \sigma^2)$, we have that all the perturbations of weights $\|\Delta.\|$ are bounded by $\sigma\sqrt{2h\ln(24h)} : \omega$ with probability larger than $1/2$.

Since the only difference between OptEqs and monDEQ [51] is the weight parameterization, our reformulate OptEqs is parameterized by $\mathbf{W}_s = \mathbf{W}^\top\mathbf{W}$ while the monDEQ's weight parameter is parameterized by a series of weights $\mathbf{W}_{mondeq} = \mathbf{I} + \mathbf{A} + \mathbf{A}^\top + \mathbf{B} + \mathbf{B}^\top$. Therefore, the pertubation in $\|\Delta_{\mathbf{W}_s}\|$ is different from former analysis [38]. We have that:

$$\|\Delta_{\mathbf{W}_s}\|_2 = \|\Delta_{\mathbf{W}}^\top\mathbf{W} + \mathbf{W}\Delta_{\mathbf{W}}^\top\|_2 \leq 2\omega\mu \tag{29}$$

Then using the above lemma, we have that for all $\mathbf{x}$ with probability at least $1/2$.

$$\|\overline{f}_{opteq}^c(\mathbf{x}) - f_{opteq}^c(\mathbf{x})\|_2 \leq \|\overline{\mathbf{W}}_c\overline{f}_{opteq}(\mathbf{x}) + \overline{\mathbf{b}}_c - \mathbf{W}_c f_{opteq}^c(\mathbf{x}) - \mathbf{b}\|_2$$
$$\leq \frac{2\mu^2\omega(B+1)}{(1-m)^2} + \frac{2\mu\omega(B+1)}{1-m} + \omega \tag{30}$$

Setting $\sigma = \frac{\eta(1-m^2)}{4\sqrt{2h\ln(24h)}(2\mu^3(B+1)+2(1-m)\mu+(1-m)^2)}$ will make the above perturbation less than $\frac{\eta}{4}$. Then we have,

$$KL(\mathbf{W}. + \Delta_{\mathbf{W}.}|P) \leq \frac{\mathbf{B}_W}{2\sigma^2} = \frac{16h\ln(24h)(2\mu^3(B+1) + 2(1-m)\mu(B+1) + (1-m)^2)^2}{\eta^2(1-m)^4}\mathcal{B}_W \tag{31}$$

With the same choice of $\beta$'s bound like former work [38],

$$\frac{\eta(1-m)}{2(B+1)} \leq \beta \leq \frac{\eta(1-m)\sqrt{M}}{2(B+1)}, \tag{32}$$

we can finally get the upper bound as our OptEqs bound as our proposition shows.

The difference between GEQ and OptEqs is that GEQ's can be viewed as multiplying scaler $\beta = e^{-\gamma\|\mathbf{U}\mathbf{x}+\mathbf{b}-\mathbf{W}\mathbf{z}\|_2^2}$ with depend on $\mathbf{x}$ since $\mathbf{z}$ is also depended on $\mathbf{x}$. Setting $\beta_{\max} = \max_{\mathbf{x}\in\mathcal{D}}\beta(\mathbf{x}) < 1$ and $\beta_{\min} = \min_{\mathbf{x}\in\mathcal{D}}\beta(\mathbf{x}) > c$. Assuming $\beta$ changes a little with respect to the small perturbations on weights, we have:

$$\mathbf{z}^*(\mathbf{W}, \mathbf{U}, \mathbf{b}) = \sigma(-\beta\mathbf{W}_s\mathbf{z}^{(i)} + \beta\mathbf{W}(\mathbf{U}\mathbf{x} + \mathbf{b}))$$
$$\|\mathbf{z}^*(\overline{\mathbf{W}}, \overline{\mathbf{U}}, \overline{\mathbf{b}})\|_2 \leq \frac{\beta_{\max}\mu\|\overline{\mathbf{W}}_s - \mathbf{W}_s\|_2\|\mathbf{U}\mathbf{x}+\mathbf{b}\|_2}{(1-\beta_{\max}m)^2} + \frac{\beta_{\max}(\|(\overline{\mathbf{W}\mathbf{U}} - \mathbf{W}\mathbf{U})\mathbf{x}\|_2 + \|\overline{\mathbf{W}\mathbf{b}} - \mathbf{W}\mathbf{b}\|_2)}{1-\beta_{\max}m} \tag{33}$$

With the same setting as above OptEqs, we have:

$$\|(\overline{\mathbf{W}\mathbf{U}} - \mathbf{W}\mathbf{U})\mathbf{x}\|_2 = \|\Delta_{\mathbf{W}}\overline{\mathbf{U}} - \mathbf{W}\Delta_{\mathbf{U}}\mathbf{x}\|_2 \leq 2\omega\mu \quad \|(\overline{\mathbf{W}\mathbf{b}} - \mathbf{W}\mathbf{b})\mathbf{x}\|_2 = \|\Delta_{\mathbf{W}}\overline{\mathbf{b}} - \mathbf{W}\Delta_{\mathbf{b}}\mathbf{x}\|_2 \leq 2\omega\mu \tag{34}$$

Then we can obtain that:

$$\|\overline{f}_{geq}^c(\mathbf{x}) - f_{kereq}^c(\mathbf{x})\|_2 \leq \|\overline{\mathbf{W}}_c\overline{f}_{opteq}(\mathbf{x}) + \overline{\mathbf{b}}_c - \mathbf{W}_c f_{opteq}^c(\mathbf{x}) - \mathbf{b}\|_2$$
$$\leq \frac{2\beta_{\max}\mu^3\omega(B+1)}{(1-\beta_{\max}m)^2} + \frac{2\mu^2\omega\beta_{max}(B+1)}{1-\beta_{max}m} + \frac{\mu\omega(B+1)}{1-\beta_{max}m} + \omega$$
$$= \frac{(\beta_{\max}\mu)*(2\mu^2\omega(B+1))}{(1-\beta_{\max}m)^2} + \frac{(2\mu\beta_{max}+1)*(\mu\omega(B+1))}{1-\beta_{max}m} + \omega \tag{35}$$

With the same setting as above, we have:

$$KL(\mathbf{W}. + \Delta_{\mathbf{W}.}|P) \leq \frac{\mathbf{B}_W}{2\sigma^2}$$
$$= T\mathcal{B}_W, \tag{36}$$

where $T$ is defined as follows:

$$T = \frac{16h\ln(24h)(2(\beta_{\max}\mu)\mu^3(B+1) + 2(2\mu\beta_{\max}+1)(1-\beta_{\max}m)\mu(B+1) + (1-\beta_{\max}m)^2)^2}{\eta^2(1-\beta_{\max}m)^4} \tag{37}$$

then we can finally we can get the upper bound as our OptEqs bound as our proposition shows.

## A.4 Proofs for proposition 3

Lipschitz constant is the minimal constant for $f$ and $\forall \mathbf{x}, \mathbf{y}$ suits the following equation:

$$\|f(\mathbf{x}) - f(\mathbf{y})\|_2 \le L\|\mathbf{x} - \mathbf{y}\|_2. \tag{38}$$

Thereby, our analysis in Proposition 3 can be viewed as proposing an upper bound for different models. In this section, we are going to prove the Lipschitz upper bounds for our GEQ and OptEqs. First, we restate the proposition as follows:

**Proposition 3.** *If norms for the inputs and outputs are bounded by $B$, the spectral norm for the weight parameter $\mathbf{W}, \mathbf{U}$ of equilibrium models with ReLU activation are bounded by $\mu < 1$ to ensure convergence, then we have the conclusions as below:*

$$\|f_{geq}(\mathbf{x}_1) - f_{geq}(\mathbf{x}_2)\|_2 \le L_{geq}\|\mathbf{x}_1 - \mathbf{x}_2\|_2 = \frac{\beta_{\max}\mu^2 + \sqrt{\gamma}B\mu^3}{1 - \beta_{\max}\mu^2 - \sqrt{\gamma}B\mu^3}\|\mathbf{x}_1 - \mathbf{x}_2\|_2, \tag{39}$$

$$\|f_{opteq}(\mathbf{x}_1) - f_{opteq}(\mathbf{x}_2)\|_2 \le L_{opteq}\|\mathbf{x}_1 - \mathbf{x}_2\|_2 = \frac{\mu}{1 - \mu^2}\|\mathbf{x}_1 - \mathbf{x}_2\|_2, \tag{40}$$

*where $\mathbf{x}_1$ and $\mathbf{x}_2$ are input samples, $f_{geq}(\mathbf{x}.)$ and $f_{opteq}(\mathbf{x}.)$ denotes the equilibrium states for GEQ and OptEqs given input $\mathbf{x}.$, and $\beta_{max} := \max_{\mathbf{x}\in\mathcal{D}} e^{-\gamma\|\mathbf{U}\mathbf{x} - \mathbf{W}\mathbf{z}\|_2^2} < 1$.*

*Proof.* We first prove the inequality for OptEqs:

$$
\begin{aligned}
\|f_{opteq}(\mathbf{x}_1) - f_{opteq}(\mathbf{x}_2)\|_2 = \|\mathbf{z}_{\mathbf{x}_1} - \mathbf{z}_{\mathbf{x}_2}\|_2 &\le \left\|\sigma\left(\mathbf{W}^\top\mathbf{W}\mathbf{z}_{\mathbf{x}_1} + \mathbf{U}\mathbf{x}_1\right) - \sigma\left(\mathbf{W}^\top\mathbf{W}\mathbf{z}_{\mathbf{x}_2} + \mathbf{U}\mathbf{x}_2\right)\right\|_2 \\
&\le \left\|\mathbf{W}^\top\mathbf{W}(\mathbf{z}_{\mathbf{x}_1} - \mathbf{z}_{\mathbf{x}_2})\right\|_2 + \|\mathbf{U}(\mathbf{x}_1 - \mathbf{x}_2)\|_2 \\
&\le \mu^2\|\mathbf{z}_{\mathbf{x}_1} - \mathbf{z}_{\mathbf{x}_2}\|_2 + \mu\|\mathbf{x}_1 - \mathbf{x}_2\|_2,
\end{aligned}
\tag{41}
$$

where $\mathbf{z}_{\mathbf{x}_1}$ denotes the equilibrium states for OptEq with input $\mathbf{x}_1$, which means the following equation is satisfied:

$$\mathbf{z}_{\mathbf{x}_1} = \sigma\left(\mathbf{W}^\top\mathbf{W}\mathbf{z}_{\mathbf{x}_1} + \mathbf{U}\mathbf{x}_1\right). \tag{42}$$

Reformulating the equations, we can get:

$$\|f_{opteq}(\mathbf{x}_1) - f_{opteq}(\mathbf{x}_2)\|_2 \le \frac{\mu}{1 - \mu^2}\|\mathbf{x}_1 - \mathbf{x}_2\|_2. \tag{43}$$

Then for GEQ, we have:

$$
\begin{aligned}
\|\mathbf{z}_{\mathbf{x}_1} - \mathbf{z}_{\mathbf{x}_2}\|_2 &\le \left\|\sigma\left(e^{-\gamma\|\mathbf{W}\mathbf{z}_{\mathbf{x}_1} - \mathbf{U}\mathbf{x}_1\|_2^2}\left(-\mathbf{W}^\top\mathbf{W}\mathbf{z}_{\mathbf{x}_1} + \mathbf{W}^\top\mathbf{U}\mathbf{x}_1\right)\right)\right. \\
&\qquad \left. - \sigma\left(e^{-\gamma\|\mathbf{W}\mathbf{z}_{\mathbf{x}_2} - \mathbf{U}\mathbf{x}_2\|_2^2}\left(-\mathbf{W}^\top\mathbf{W}\mathbf{z}_{\mathbf{x}_2} + \mathbf{W}^\top\mathbf{U}\mathbf{x}_2\right)\right)\right\|_2 \\
&\le \beta_{\max}\left\|\mathbf{W}^\top\mathbf{W}(\mathbf{z}_{\mathbf{x}_1} - \mathbf{z}_{\mathbf{x}_2})\right\|_2 + \left\|\mathbf{W}^\top\mathbf{U}(\mathbf{x}_1 - \mathbf{x}_2)\right\|_2 + B\mu^2|\beta_{\mathbf{x}_1} - \beta_{\mathbf{x}_2}|, \\
&\le \beta_{\max}\mu^2\|\mathbf{z}_{\mathbf{x}_1} - \mathbf{z}_{\mathbf{x}_2}\|_2 + \beta_{\max}\mu^2\|\mathbf{x}_1 - \mathbf{x}_2\|_2 + B\mu^2|\beta_{\mathbf{x}_1} - \beta_{\mathbf{x}_2}|
\end{aligned}
\tag{44}
$$

where $\mathbf{z}_{\mathbf{x}_1}$ denotes the equilibrium states for GEQ with input $\mathbf{x}_1$, and $\beta_{\mathbf{x}_1}$ is defined as follows:

$$\beta_{\mathbf{x}_1} = e^{-\gamma\|\mathbf{W}\mathbf{z}_{\mathbf{x}_1} - \mathbf{U}\mathbf{x}\|_2^2}. \tag{45}$$

Then with the mean value theorem, we have the following equations:

$$
\begin{aligned}
|\beta_{\mathbf{x}_1} - \beta_{\mathbf{x}_2}| = \left|e^{-\gamma\|\mathbf{W}\mathbf{z}_{\mathbf{x}_1} - \mathbf{U}\mathbf{x}_2\|_2^2} - e^{-\gamma\|\mathbf{W}\mathbf{z}_{\mathbf{x}_1} - \mathbf{U}\mathbf{x}_2\|_2^2}\right| \\
\le \sqrt{\gamma}|\,\|\mathbf{W}\mathbf{z}_{\mathbf{x}_1} - \mathbf{U}\mathbf{x}_2\|_2 - \|\mathbf{W}\mathbf{z}_{\mathbf{x}_x} - \mathbf{U}\mathbf{x}_2\|_2\,|\cdot \max_{c\in\sqrt{\gamma}[\|\mathbf{W}\mathbf{z}_{\mathbf{x}_1} - \mathbf{U}\mathbf{x}_1\|_2, \|\mathbf{W}\mathbf{z}_{\mathbf{x}_2} - \mathbf{U}\mathbf{x}_2\|_2]} ce^{-c^2} \\
\le \sqrt{\gamma}|\,\|\mathbf{W}\mathbf{z}_{\mathbf{x}_2} - \mathbf{U}\mathbf{x}_2\|_2 - \|\mathbf{W}\mathbf{z}_{\mathbf{x}_2} - \mathbf{U}\mathbf{x}_2\|_2\,| \\
\le \sqrt{\gamma}\|\mathbf{W}(\mathbf{z}_{\mathbf{x}_1} - \mathbf{z}_{\mathbf{x}_2})\|_2 + \|\mathbf{U}(\mathbf{x}_1 - \mathbf{x}_2)\|_2 \\
\le \sqrt{\gamma}\mu\left(\|\mathbf{z}_{\mathbf{x}_1} - \mathbf{z}_{\mathbf{x}_2}\|_2 + \|\mathbf{x}_1 - \mathbf{x}_2\|_2\right)
\end{aligned}
\tag{46}
$$

Then reformulating Eqn (44), we can finally obtain our proposition as below:

$$\|\mathbf{z}_{\mathbf{x}_1} - \mathbf{z}_{\mathbf{x}_2}\|_2 = \|f_{geq}(\mathbf{x}_1) - f_{geq}(\mathbf{x}_2)\|_2 \le \frac{\beta_{\max}\mu^2 + \sqrt{\gamma}B\mu^3}{1 - \beta_{\max}\mu^2 - \sqrt{\gamma}B\mu^3}\|\mathbf{x}_1 - \mathbf{x}_2\|_2 \tag{47}$$

$\square$

## A.5 Proofs for proposition 4

In this section, we are going to prove the output similarity bounds for our GEQ and OptEqs. First, we restate the proposition as follows:

**Proposition 4.** *If norms for the inputs and outputs are bounded by $B$, the spectral norm for the weight parameter $\mathbf{W}$ of equilibrium models with ReLU activation are bounded by $\mu < 1$ to ensure the convergence, and each row in $\mathbf{U}$ obeys the spherical Gaussian distributions $\mathcal{N}(0, \mathbb{E}[U_i^2]\mathbf{I})$. Then we have the following conclusions for the expectation of the output similarity for GEQ and OptEqs with respect to input $\mathbf{x}_1, \mathbf{x}_2$ as follows,*

$$\kappa_{geq}(\mathbf{x}_1, \mathbf{x}_2) \leq \overline{\kappa}_{geq} = \frac{\mu^2 D e^{-\frac{\gamma}{4}(\sigma_{\min}(\mathbf{U})^2 \|\mathbf{x}_1 - \mathbf{x}_2\|_2^2)} \mathbb{E}[U_i^2] \|\mathbf{x}_1\| \|\mathbf{x}_2\| (\sin\theta_0 + (\pi - \theta_0)\cos\theta_0)}{2\pi(1 - \beta_{\max}\mu^2)^2}, \tag{48}$$

$$\kappa_{opteq}(\mathbf{x}_1, \mathbf{x}_2) \leq \overline{\kappa}_{opteq} = \frac{\mathbb{E}[U_i^2] \|\mathbf{x}_1\| \|\mathbf{x}_2\| (\sin\theta_0 + (\pi - \theta_0)\cos\theta_0)}{2\pi(1 - \mu^2)^2}, \tag{49}$$

*where $\mathbf{x}_1$ and $\mathbf{x}_2$ are input samples, $D = e^{\gamma B \|\mathbf{W}\|_2^2}$ and $\beta_{max} := \max_{\mathbf{x} \in \mathcal{D}} e^{-\gamma \|\mathbf{U}\mathbf{x} - \mathbf{W}\mathbf{z}\|_2^2} < 1$. $\theta_0 = \cos^{-1}(\frac{\langle \mathbf{x}_1, \mathbf{x}_2 \rangle}{\|\mathbf{x}_1\| \|\mathbf{x}_2\|})$ is defined as the angle between the samples.*

*Proof.* Before the proof, we first introduce the following lemma:

**Lemma 3.** *[48] If $\mathbf{U}$ obeys the spherical Gaussian distributions of variance $\mathbb{E}[U_i^2]$ and mean 0, then the expectation of the Similarity for the one-layer Neural Network $\sigma(\mathbf{U}\mathbf{x})$ is:*

$$\kappa_{NN}(\mathbf{x}_1, \mathbf{x}_2) = \frac{\mathbb{E}[U_i^2] \|\mathbf{x}_1\| \|\mathbf{x}_2\|}{2\pi} (\sin\theta_0 + (\pi - \theta_0)\cos\theta_0) \tag{50}$$

*where $\theta_0 = \cos^{-1}(\frac{\langle \mathbf{x}, \mathbf{y} \rangle}{\|\mathbf{x}\| \|\mathbf{y}\|})$.*

Letting $m := \|\mathbf{W}_s\|_2 = \|\mathbf{W}^\top \mathbf{W}\|_2 < 1$ and $\mu := \|\mathbf{W}\|_2 < 1$ as our assumptions demonstrate and neglecting the bias $\mathbf{b}$ for convenience. Then for our reformulated OptEqs, we have:

$$\begin{aligned}
\mathbf{z}_{\mathbf{x}_1}^{*\top} \mathbf{z}_{\mathbf{x}_2} &\leq \sigma(\mathbf{W}_s \mathbf{z}_{\mathbf{x}_1})^\top \mathbf{z}_{\mathbf{x}_2} + \sigma(\mathbf{U}\mathbf{x}_1)^\top \mathbf{z}_{\mathbf{x}_2} \\
&\leq \mu^2 \sigma(\mathbf{z}_{\mathbf{x}_1})^\top \mathbf{z}_{\mathbf{x}_2} + \sigma(\mathbf{U}\mathbf{x}_1)^\top \mathbf{z}_{\mathbf{x}_2}
\end{aligned} \tag{51}$$

And we have

$$\begin{aligned}
\sigma(\mathbf{U}\mathbf{x}_1)^\top \mathbf{z}_{\mathbf{x}_2} &\leq \sigma(\mathbf{U}\mathbf{x}_1)^\top \sigma(\mathbf{W}\mathbf{z}_{\mathbf{x}_2}) + \sigma(\mathbf{U}\mathbf{x}_1)^\top \sigma(\mathbf{U}\mathbf{x}_2) \\
&\leq \mu^2 \sigma(\mathbf{U}\mathbf{x}_1)^\top \mathbf{z}_{\mathbf{x}_2} + \sigma(\mathbf{U}\mathbf{x}_1)^\top \sigma(\mathbf{U}\mathbf{x}_2)
\end{aligned} \tag{52}$$

Then

$$\begin{aligned}
\sigma(\mathbf{U}\mathbf{x}_1)^\top \mathbf{z}_{\mathbf{x}_2} &\leq \frac{\sigma(\mathbf{U}\mathbf{x}_1)^\top \sigma(\mathbf{U}\mathbf{x}_2)}{1 - \mu^2} \\
\mathbf{z}_{\mathbf{x}_1}^{*\top} \mathbf{z}_{\mathbf{x}_2} &\leq \frac{\sigma(\mathbf{U}\mathbf{x}_1)^\top \sigma(\mathbf{U}\mathbf{x}_2)}{(1 - \mu^2)^2}
\end{aligned} \tag{53}$$

Therefore, we can conclude

$$\kappa_{opteq}(\mathbf{x}_1, \mathbf{x}_2) \leq \frac{\mathbb{E}[U_i^2] \|\mathbf{x}_1\| \|\mathbf{x}_2\| (\sin\theta_0 + (\pi - \theta_0)\cos\theta_0)}{2\pi(1 - \mu^2)^2} \tag{54}$$

For GEQ, we set $\beta_{\mathbf{x}} = e^{-\gamma \|\mathbf{U}\mathbf{x} - \mathbf{W}\mathbf{z}\|_2^2}$ and $\beta_{max} := \max_{\mathbf{x} \in \mathcal{D}} e^{-\gamma \|\mathbf{U}\mathbf{x} - \mathbf{W}\mathbf{z}\|_2^2} < 1$

$$\begin{aligned}
\mathbf{z}_{\mathbf{x}_1}^{*\top} \mathbf{z}_{\mathbf{x}_2} &\leq \beta_{\max} \sigma(\mathbf{W}_s \mathbf{z}_{\mathbf{x}_1})^\top \mathbf{z}_{\mathbf{x}_2} + \beta_{\mathbf{x}} \|\mathbf{W}\|_2 \sigma(\mathbf{U}\mathbf{x}_1)^\top \mathbf{z}_{\mathbf{x}_2} \\
&\leq \beta_{\max} \mu^2 \sigma(\mathbf{z}_{\mathbf{x}_1})^\top \mathbf{z}_{\mathbf{x}_2} + \beta_{\mathbf{x}} \mu \sigma(\mathbf{U}\mathbf{x}_1)^\top \mathbf{z}_{\mathbf{x}_2}
\end{aligned} \tag{55}$$

Like OptEqs, we obtain the following equations for GEQ:

$$\mathbf{z}_{\mathbf{x}_1}^{*\top} \mathbf{z}_{\mathbf{x}_2} \leq \frac{\mu^2 \beta_{\mathbf{x}_1} \beta_{\mathbf{x}_2} \sigma(\mathbf{U}\mathbf{x}_1)^\top \sigma(\mathbf{U}\mathbf{x}_2)}{(1 - \beta_{\max}\mu^2)^2} \tag{56}$$

Therefore,

$$\kappa_{geq}(\mathbf{x}_1, \mathbf{y}_2) \leq \frac{\mu^2 \beta_{\mathbf{x}_1} \beta_{\mathbf{x}_2} \mathbb{E}[U_i^2] \|\mathbf{x}_1\|_2 \|\mathbf{x}_2\|_2 \left(\sin\theta_0 + (\pi - \theta_0)\cos\theta_0\right)}{2\pi(1 - \beta_{\max}\mu^2)^2} \tag{57}$$

And we have,

$$\begin{aligned} \beta_{\mathbf{x}_1} \beta_{\mathbf{x}_2} &= e^{-\gamma(\|\mathbf{W}\mathbf{z}_{\mathbf{x}_1} - \mathbf{U}\mathbf{x}_1\|_2^2 + \|\mathbf{W}\mathbf{z}_{\mathbf{x}_2} - \mathbf{U}\mathbf{x}_2\|_2^2)} \\ &\leq e^{-\gamma/2(\|\mathbf{W}(\mathbf{z}_{\mathbf{x}_1} - \mathbf{z}_{\mathbf{x}_2}) - \mathbf{U}(\mathbf{x}_1 - \mathbf{x}_2)\|_2^2)} \\ &\leq D e^{-\frac{\gamma}{4}(\sigma_{\min}(\mathbf{U})^2 \|\mathbf{x}_1 - \mathbf{x}_2\|_2^2)}, \end{aligned} \tag{58}$$

where $D = e^{\gamma\|\mathbf{W}\|_2^2 B}$, the latter two inequality is acquired by Jensen's inequality. Then we have,

$$\kappa_{geq}(\mathbf{x}_1, \mathbf{x}_2) \leq \frac{\mu^2 D e^{-\frac{\gamma}{4}(\sigma_{\min}(\mathbf{U})^2 \|\mathbf{x}_1 - \mathbf{x}_2\|_2^2)} \mathbb{E}[U_i^2] \|\mathbf{x}_1\|_2 \|\mathbf{x}_2\|_2 \left(\sin\theta_0 + (\pi - \theta_0)\cos\theta_0\right)}{2\pi(1 - \beta_{\max}\mu^2)^2} \tag{59}$$

$\square$

## A.6 Experiment Settings

### A.6.1 Experiments on CIFAR

For our GEQ, we parallel 6 branches with each branch taking the scale of $32, 16, 8, 8, 4, 4$ and using the average fusion method for branches' fusion. The output channels for 6 branches are all $256$ or $320$ but the mid-channel number(output channel for weight $\mathbf{U}$ and $\mathbf{W}$) for the six branches are $64, 128, 128, 128, 256, 256$ or $80, 160, 160, 160, 320, 320$ with patch size 2 and c splitting is 8. And the inner MLP inner $\mathbf{W}_h$ output 64 hidden dimension for each patch. We use the SGD [23] optimizer with momentum and step learning rate schedule for all the models. We also use RandomAug for all the models for comparison.

### A.6.2 Experiments on ImageNette and ImageNet-100

We take the input scale as $256$ for all models. For our GEQ, we parallel 6 branches with each branch taking the scale of $64, 32, 16, 16, 8, 8$ after two downsampling convolutions and using the average fusion method for branches' fusion. The output channels for 6 branches are all $256$ or $384$ but the mid-channel number(output channel for weight $\mathbf{U}$ and $\mathbf{W}$) for the six branches are $32, 64, 128, 128, 256, 256$ or $48, 96, 192, 192, 384, 384$ with patch size 2 and c splitting is 8. And the inner MLP inner $\mathbf{W}_h$ output 128 hidden dimension for each patch. We use the SGD optimizer with momentum and step learning rate schedule for all the models. We also use RandomAug for all the models for comparison.

