# OpenReview forum: "GEQ: Gaussian Kernel Inspired Equilibrium Models"
_NeurIPS.cc/2023/Conference — NeurIPS 2023 poster_

### Official Review · Reviewer_sG64 · 2023-06-22

**Soundness:** 3 good
**Presentation:** 3 good
**Contribution:** 3 good
**Rating:** 6
**Confidence:** 3

**Summary:**

This paper makes the observation that conventional DEQ architectures effectively use linear kernels in their feature extraction, and the proposed method in this paper is to use Gaussian Kernels instead of linear kernels. Then, the paper makes the observation that adding Gaussian Kernels to DEQ models produce an architecture that is effectively infinite deep and infinitely wide. The paper then proposes patch splitting, a technique that makes the proposed method enjoy better performance, and shows the effectiveness of the proposed method on CIFAR10, CIFAR100, and saliency map datasets.

**Strengths:**

This paper is clearly written. In terms of contribution, this paper makes the observation that the proposed method is effectively infinitely deep and infinitely wide, which I find to be an interesting thread. Finally, this paper contains convergence bounds and stability analysis, training technique (patch splitting), as well as numerous satisfactory experiment results, and therefore I believe that the proposed method is sufficiently interesting.

**Weaknesses:**

I believe this paper would benefit more from discussion about the proposed model. For example, now that we know the proposed model is infinitely deep and infinitely wide, what can we say about its connection to NTK? Can this model further some of the NTK analysis? In addition, can this model inspire potential work on Gaussian processes?

**Questions:**

1. Similar to the weakness section---I am curious whether the proposed model can act as an effective model for large neural nets in general, the same way that the NTK regime approximates large width networks? Can this model be representative of large language models, given its infinite depth and infinite width capacity?
2. Does the Gaussian Kernel in the proposed model have any natural use cases in practice (that linear kernels, polynomial kernels, etc don't have)?

**Limitations:**

There are no negative societal impacts of this work

---

> ### Author Rebuttal · Authors · 2023-08-10
>
> Thanks for your comments. The followings are our responses to your concerns.
>
> # 1. About the relationship between GEQ and NTK.
>
> Both GEQ and NTK share a fundamental likeness: they align with the concept of infinite-width models. And we also reference some theoretical results in NTK and Gaussian process models for our analysis. However, our GEQ doesn’t rely on some unrealistic weight distribution. Therefore, the training process for GEQ is much easier, which leads to its superior performance over NTK. Consequently, GEQ's capabilities may have the potential to instigate innovative approaches to NTK model development, making such models more useful and realistic.
>
> Anticipating the future trajectory of GEQ and NTK analyses, we posit that the kernel mechanism intrinsic to GEQ could serve as a wellspring for novel architectural designs within the NTK framework. An intriguing way to explore involves integrating the strengths of both methodologies. This entails harnessing the intra-sample kernel calculation akin to GEQ, which we think is one of the key reasons that GEQ’s performance is better than NTK. Apart from that, the new model should also leverage NTK's global kernel function computed over the entire training dataset. This hybrid approach has the promise of delivering a robust performance akin to GEQ and simultaneously facilitating the acquisition of dataset-specific properties, such as out-of-distribution (OOD) generalization and fairness considerations.
>
> Shifting the focus to the advancement of Gaussian process models, GEQ imparts a valuable insight—equilibrium models could constitute a compelling route for expanding Gaussian process models from their non-parametric state to a parameterized one. By embracing this equilibrium framework, these models stand to achieve heightened performance levels and open avenues for significant enhancements.
>
> # 2. About the relationship between GEQ and large neural networks.
>
> Regarding the connection between our GEQ and large neural networks, it is worth noting that our GEQ possesses attributes akin to those of a weight-tied, large deep neural network (as illustrated in [1]) accompanied by attention modules. This resemblance stems from the utilization of exponential terms As such, our GEQ holds promise in providing theoretical insights into the realm of deep and expansive neural networks enriched with attention modules.
>
> Turning to the potential interplay between our GEQ and current large models, such as Large Language Models or Multi-Modal Models, we think our GEQ shows a closer connection with multi-modal models instead of LLM because our GEQ-induced attentive module can be viewed as cross-attention rather than self-attention mechanisms.
>
> Furthermore, our GEQ model itself can scale to large and show better performance than large deep ResNets. For example, on ImageNet:
>
> | Model | Model Size | Test Acc |
> | --- | --- | --- |
> | ResNet-18 | $13$M | $70.2\%$ |
> | ResNet-50 | $26$M | $75.1\%$ |
> | GEQ | $16$M | $\mathbf{75.9}\%$ |
>
> The results underscore a notable observation: when our GEQ is scaled up to larger models, it demonstrates superior performance compared to deep ResNets, even while employing fewer parameters, particularly evident when evaluated on larger ImageNet datasets.
>
> # 3. About "Does the Gaussian Kernel in the proposed model have any natural use cases in practice (that linear kernels, polynomial kernels, etc don't have)?"
>
> Equilibrium models endowed with linear kernels can be conceptualized as akin to weight-tied ResNets or analogous models, as exemplified in [1]. Consequently, our GEQ, employing a Gaussian kernel, can be likened to weight-tied ResNets, augmented by attention modules like SE-Net [2], Graph Attention Network[3] or etc.
>
> [1] Deep equilibrium Models
>
> [2] Squeeze-and-Excitation Networks
>
> [3] Graph Attention Networks

---

> > ### Comment · Reviewer_sG64 · 2023-08-21
> >
> > Many thanks to the authors for taking time to address my questions. I have also read carefully over the feedback of other reviewers. I would leave the question of whether this paper should be accepted for the AC to decide.

---

### Official Review · Reviewer_CdMn · 2023-06-30

**Soundness:** 1 poor
**Presentation:** 2 fair
**Contribution:** 2 fair
**Rating:** 4
**Confidence:** 3

**Summary:**


The authors introduce a deep equilibrium model (6) which ostensibly solves an optimisation problem (5) involving a Gaussian (squared exponential) kernel. The authors describe some possible theoretical properties of the new model, and benchmark it on some problems against some related optimisation-inspired DEQs and resnets.

**Strengths:**

- The idea of deriving DEQ models from optimisation problems is nice, and fits in well with existing literature.

**Weaknesses:**

Unfortunately, there appear to be many issues with the mathematical results in this paper. I do not believe they are sound. Some of them could potentially be fixable, but I am not certain.

- **Important** The use of the function $f$ is not clear.
    - Can you elaborate on the definition of $f$? On the second page, you say "$f$ is a positive indicator function that induces to commonly used ReLU activation" What does "induces to" mean? What exactly is your definition of indicator function? One definition is that given a set $A$, $f_A(z) = 1$ if $z \in A$ and zero otherwise. This does not seem to reconcile with your definition, though. Perhaps $f$ takes a value of $1$ if $z$ is greater than 0 (elementwise), and zero otherwise? What happens at the value of zero? What is $f$'s (weak) derivative? How is it related to the ReLU?
    - After equation (3), you mention the first order stationarity condition. Is $f$ here the indicator function? Isn't this function not everywhere differentiable? Even if the derivative were defined, how do you know that $\nabla G = 0$ implies a minima?
    - How is a ReLU activation in (4) obtained by differentiation? I could see that the derivative of $relu(a)^2$ with respect to $a$ would be $relu(a)$, but not sure how to obtain it otherwise?
    - Proposition 1. What is the meaning of the symbol $\partial f$? How is $(1+\partial f)^{-1}$ related to the ReLU function?
- Proposition 1 is not precise. The inner product is not defined, which is quite important because $\Phi$ lives in some kind of Hilbert space (informally being referred to as $\mathbb{R}^\infty$). Can you clarify what $\infty$-dimensional space means, and what $\langle \cdot, \cdot \rangle $ means?
- Proposition 3. Isn't the left hand side of equation (11) zero? This would be a trivial bound, I would have thought.
- (Minor) Proof of proposition 3. Aren't all the first lines in equation (41) equality, by definition of fixed point?
- Proposition 4. Can you elaborate on why it is helpful to exhibit a smaller output similarity for dissimilar samples? One can trivially do this for any kernel, by simply rescaling the kernel by a constant factor.


**Questions:**

Minor (*not* a reason to reject):
- Section title for section3 is missing an "I".
- The titles and labels in Figure 2 are too small to read.
- Why are only Gaussian kernels considered?
- Many works have pointed out potential issues with using saliency maps such as GradCAM to try and gain insight into neural network predictions. E.g. see Cynthia Rudin's work on explainability versus interpretability. The saliency map visualisation seems to come out of nowhere (not being mentioned anywhere earlier in the paper). Is there a reason why include it in this paper?

**Limitations:**

I do not see any explicit mention of limitations in this work. That being said, I may have missed some, if they are somewhere in the text.

---

> ### Author Rebuttal · Authors · 2023-08-10
>
> Thanks for your comments. The followings are our responses to your concerns.
>
> # 1. About $f$:
>
> ## 1) The definition and differentiability of $f$:
>
> The definition of $f$ is $f(x) = 1$ when  $x \geq 0$ while $f(x)=\infty$ when $x<0$.  And $f$ is sub-differentiable and  $\partial f$ here means the sub-gradients "[subgradients](https://web.stanford.edu/class/ee364b/lectures/subgradients_notes.pdf)". In our paper, we set $\partial f=0$ at $x=0$  for convenience since $0\in \partial f$. We'll rewrite them to make it more clear in the future version.
>
> ## 2) About the stationary condition:
>
> Although $f$ and $G$ are not differentiable everywhere, the stationary condition can also be used with $G$’s sub-gradient, i.e., $0 \in \partial G$.  More details in "[subgradients](https://web.stanford.edu/class/ee364b/lectures/subgradients_notes.pdf)". We use $0=\nabla G$  in our paper for convenience since we set $\partial f=0$ at $x=0$, and our equilibrium also suits the above condition $0 \in \partial G$. Therefore, our conclusion is correct although the formulation is a little non-rigorousness. We'll rewrite them to make it more clear in the future version.
>
> ## 3) About the derivative of the ReLU function.
>
> We take the following optimization problem as an example. When you solve the following optimization problem $argmin_{x\geq0} \frac{1}{2}||x-y||^2$, you can first reformulate the optimization problem as  $argmin_{x} \frac{1}{2}||x-y||^2 +f(x)$ where $f$ is defined as above since two optimization problems have the same optimal point.
>
> Then the optimal solution can be achieved by calculating the problems subgradient $0 =  (1+f)(x) -y$ and then we can get $x=(1+\partial f)^{-1}(y)$ as the optimal solution. And we can also get the original problem's optimal solution as $x=0$ when $y<0$ because $f(x)=\infty$ when $x<0$ and $x=y$ when $y\geq 0$. Therefore, $(1+\partial f)^{-1}(y) = max(y,0)$ which is ReLU and it also calls the proximal operator of $f$.
>
> More details in "[Proximal Operators](https://www.math.cuhk.edu.hk/course_builder/1920/math4230/Note10.pdf)".
>
> # 2. About inner-production, infinity dimension spaces’ definition in Proposition 1.
>
> The inner product in our paper is$\langle \mathbf{x}，\mathbf{y} \rangle = \mathbf{x}^\top\mathbf{y}$, which you can find in "[Inner Product](https://mathworld.wolfram.com/InnerProduct.html)". And the $\infty-$dimensional space means the vector $\mathbf{v}$ in such space has infinite dimension which means $\mathbf{v}\in\mathbb{R}^{\infty}$.
>
> # 3. About Proposition 3.
>
> Although the left lower bound for Eqn(11) is 0, proposition 3 is non-trivial. A tighter Lipschitz upper bound is important because it can make the model more stable. Due to this reason, there are many works (like [1]) working on it. The upper bound can lead researchers to explore the model's most unstable performance in the worst cases. A smaller upper bound means the model is stable and can perform well on many perturbed or corrupted inputs. Thereby, proposition 3's upper bounds try to state the stability of our GEQ is better, which is also confirmed in our experiments (The second experiment in Section 4.3).
>
> As for the proof in the Appendix, the inequality in Eqn(41) is a typo and I'll correct it in the future version. But it won't influence the final results.
>
> # 4. About Proposition 4.
>
> Small output similarity for dissimilar samples will make the classification easier. For example, a dataset is easy to classify when all samples belong to the same class cluster together(similar samples should have similar representation) while the distances between cluster centers are large (dissimilar samples’ representation should be different).
>
> Therefore, simply scaling cannot make the classification performance better because it will scale all sample's similarities even if they are similar, which will make samples hard to classify.
>
> Apart from that, simply scaling with a scaler may also make OptEqs' outputs trivial. Because  simply scaling by a hyper-parameter $\eta$  to make the equilibrium function become $\mathbf{z}^* =  \sigma\left(\eta\left(\mathbf{W}^\top\mathbf{W}\mathbf{z}^*+ \mathbf{U}\mathbf{x}+\mathbf{b}\right)\right)$ can be considered as reformulating its hidden optimization problem to the following form:
>
> $\min_{\mathbf{z}}\left[\mathbf{1}^\top
> f(\mathbf{z})+ \frac{1}{2}\|\mathbf{z}\|_2^2-\eta \left \langle \mathbf{U}\mathbf{x}+\mathbf{b},\mathbf{z} \right\rangle
> -\eta \frac{1}{2}\|\mathbf{W}\mathbf{z}\|_2^2 \right].$
>
> If $\eta$  becomes too small, the final output will be more likely to be just optimizing:
>
> $\min_{\mathbf{z}}\left[\mathbf{1}^\top
> f(\mathbf{z})+ \frac{1}{2}\|\mathbf{z}\|_2^2\right]$
>
> which means the equilibrium model's final output $\mathbf{z}$ has no relationship with the original input. Therefore, the model’s outputs are useless.
>
> # 5. About "why only consider Gaussian Kernels".
>
> We not only consider Gaussian kernels. We have tried different non-linear kernels like polynomial, sigmoid, and Gaussian in Appendix A.1. However, the empirical results show that the Gaussian kernel is the best. Apart from that, gaussian kernels can also let us analyze the infinite-wide equilibrium model's properties while others cannot. Thereby, we mainly explore GEQ’s theoretical and empirical advantages.
>
> Due to the space limits, I cannot list the results here. However, you can find them in our Appendix or answers for Reviewer 8r48’s problem 3.
>
> # 6. About GradCAM.
>
> Here we use GradCAM to make readers more clear that GEQs with its induced attentive modules (as illustrated in Section 3.5) will make models focus on more semantic-related regions. We choose GradCAM here as it is an easy and widely used way to show. GradCAM is only an auxiliary technique since we have already shown our advantages with theoretical analysis and empirical comparisons.
>
> # 7. About some typos and small captions.
>
> We'll correct them in the future version.
>
> [1] Estimating Lipschitz constants of monotone deep equilibrium models

---

> > ### Comment · Area_Chair_XgmB · 2023-08-10
> > **Regarding eq 11**
> >
> > I want to clarify a point regarding eq.11 to avoid a misunderstanding: eq.11 in the paper is written as $\|f_{geq}(x_1) - f_{geq}(x_1)\|\leq L_{geq}\|x_1 - x_2\|$. The left-hand side here is clearly 0 since both terms are the same -- which is what I think the reviewer is referring to -- but I'm pretty sure that this is simply a typo and that the authors meant to write $\|f_{geq}(x_1) - f_{geq}(x_2)\|\leq L_{geq}\|x_1 - x_2\|$. Is this correct?
> >
> > Thanks,
> > AC

---

> > > ### Author Response · Authors · 2023-08-10
> > > **Comments to AC:**
> > >
> > > Thanks for pointing out. I misunderstand the reviewer CdMn’s question but now I have raised a new comment for clarification. And I’ll correct the typos in the following version.

---

> > ### Author Response · Authors · 2023-08-10
> > **To Reviewer CdMn:**
> >
> > There is a typo in Eqn(11), the correct formulation for Eqn(11) is:
> >
> >
> > $\left\| f_{geq}  (\mathbf{x}_1) - f_g (\mathbf{x}_2) \right\|_2 $
> >
> > $\leq L_{geq} \| \mathbf{x}_1 - \mathbf{x}_2\|_2$
> >
> > $= \frac{\beta_{\max}\mu^2 + \sqrt{\gamma}B\mu^3}{1-\beta_{\max}\mu^2 - \sqrt{\gamma}B\mu^3}\|\mathbf{x}_1-\mathbf{x}_2\|_2$
> >
> > Here $f_g$ denotes $f_{geq}$ due to OpenReview's bug for tex. I’ll correct them in the following version.

---

> > ### Comment · Reviewer_CdMn · 2023-08-14
> >
> > Thanks for getting back to me.
> >
> > Thanks for mentioning that you are using subgradients. As far as I can tell, subradients are not mentioned at all in the original manuscript, so it would be great if you could update the paper so that it explains that you are using subgradients rather than some other weak notion of a derivative.
> >
> > Thanks for clarifying about the definition of $f$ and ReLU. It is clear now.
> >
> > Thanks to the AC and the authors for clarifying the trivial lower bound.
> >
> > Can you clarify the following point? A local minimum (or maximum) might imply that $0 \in \partial f$. But other points can also have $0 \in \partial f$. How do you know this characterises the minima?

---

> > > ### Author Response · Authors · 2023-08-14
> > >
> > > Thank you for your response. In the upcoming version, we will denote $\partial f$ to represent the subgradient.
> > >
> > > About your concerns on “the equilibrium point is a local minimum or stationary point”, it depends on the convexity of the model’s hidden optimization problem. However, based on our configuration, we can claim that the equilibrium point in our paper is the local minimum.
> > >
> > > Firstly,  we take OptEqs as an example:
> > >
> > > $\min_{\mathbf{z}}\left[\mathbf{1}^\top
> > >   f(\mathbf{z})+ \frac{1}{2}\|\mathbf{z}\|_2^2-\left \langle \mathbf{U}\mathbf{x}+\mathbf{b},\mathbf{z} \right\rangle
> > > -\frac{1}{2}\|\mathbf{W}\mathbf{z}\|_2^2 \right]$,
> > >
> > > which can be rewritten as
> > >
> > > $\min_{\mathbf{z}\geq0}\left[ \frac{1}{2}\|\mathbf{z}\|_2^2-\left \langle \mathbf{U}\mathbf{x}+\mathbf{b},\mathbf{z} \right\rangle
> > > -\frac{1}{2}\|\mathbf{W}\mathbf{z}\|_2^2 \right]$.
> > >
> > > Then whether the equilibrium point is a stationary point or a local minimal depends on the convexity of the following part:
> > >
> > > $\frac{1}{2}\|\mathbf{z}\|_2^2-\left \langle \mathbf{U}\mathbf{x}+\mathbf{b},\mathbf{z} \right\rangle
> > > -\frac{1}{2}\|\mathbf{W}\mathbf{z}\|_2^2$.
> > >
> > > To explore the convexity, we calculate the second-order derivative (Hessian) for the above equation:
> > >
> > > $\mathbf{I}-\mathbf{W}^\top\mathbf{W}$
> > >
> > > Since all equilibrium models will use a normalization layer to ensure $\|\mathbf{W}\|_2<1$ to ensure their solutions are unique (we also state in Proposition 2,3,4), the Hessian is positive definite, which means $\mathbf{I}-\mathbf{W}^\top\mathbf{W}>0$. Thereby, the point $0\in \partial G$ can only be local minima.
> > >
> > > As for our GEQ, its formulation is a little complicated. For convenience, we will take the 1-dim case as an example where $w$ denotes $\mathbf{W}$, $u$ denotes $\mathbf{U}$, $z,x,b$ denotes $\mathbf{z},\mathbf{x},\mathbf{b}$. And the optimization problem becomes:
> > >
> > > $\min_{z>0}\left[\frac{1}{2}z^2 -\frac{1}{2\gamma}e^{-\gamma(wz-ux-b)^2}\right]$
> > >
> > > Then the second-order derivative is :
> > >
> > > $1-w^2e^{-\gamma n^2} (2\gamma n^2-1)$
> > >
> > > where $n=wz-ux-b$, since $e^{-\gamma n^2} (2\gamma n^2-1)<1$ for any $n\in\mathbb{R}$ and $w<1$. One can see that the second-order derivative is also positive. Hence it is also convex and the point $0\in \partial G$ can only be local minima. And this conclusion can also be proved when $\mathbf{z}$ is a vector.
> > >
> > > From the above analysis, one can see that the equilibrium points in our GEQ are the local minima.

---

> > > > ### Comment · Reviewer_CdMn · 2023-08-16
> > > >
> > > > The authors have done an excellent job in answering my questions and providing a definition of $f$, weak derivatives, etc. Therefore, I accordingly update my score. While I am still on the reject side of borderline, I will not strongly disagree with other reviewers who would like to see the paper accepted.

---

### Official Review · Reviewer_p1Jk · 2023-07-04

**Soundness:** 3 good
**Presentation:** 3 good
**Contribution:** 3 good
**Rating:** 7
**Confidence:** 3

**Summary:**

The presented work proposed a new equilibrium model by replacing the linear term in the original formulation into a Gaussian kernel. Analysis and experiments are performed to illustrate the superiority of the proposed model in terms of expressivity, generalization and stability, etc.

**Strengths:**

- The stability and infinite-width equavalence analysis appears to be novel. I didn't see similar analysis for other equilibrium models.
- The exploration of matching the performace of equilibrium models and traditional deep learning models is a meaningful direction. The experiments verified the effectiveness of the proposed model.

-----
I have decided to raise my rating to this paper to a clear accept because all of my concerns are well address by rebuttal.

**Weaknesses:**

- In my understanding, the modification made by the presented work is seentially replacing the linear term in OptEqs with an exponential term, which will result in an attention-like module in the forward layer. However, the idea of incorporating an exponential term into the optimization objective and unfolding it into an attention mechanism may not be considered entirely novel. For example [1] also used an exponential term and derived an attention layer from the exponential term, despite their model is not an equilibrium model and they didn't explain it as a Gaussian kernel.
- According to the Section A.2, the forward process is calculated directly by Eq.(6). However, the exponential term in Eq.(6) looks dangerous, as it is widely known that exponential function often leads to numerical instability in computations. I would expect there to be a normalization term (which leads to a softmax-like term) in Eq.(6).
- Some expressions are vague, for example it's not clear what does "feature extraction term" mean at line 48.

A minor issue: In Proposition 1 there's a term $\sqrt{{\boldsymbol{2}} \gamma} {\boldsymbol{\Phi}}_{\boldsymbol{{W}}}^{({\boldsymbol{1}})}$, what do the bolded ${\boldsymbol{2}}, {\boldsymbol{\Phi}}$ and superscript $({\boldsymbol{1}})$ mean? I guess they shouldn't be bolded and it is a typo?

[1] Transformers from an Optimization Perspective

**Questions:**

- I wonder if we can say the expressive power of GEQ is strictly stronger than the original form of OptEqs (in Eq.(3) and Eq.(4)) as the authors proved GEQ can be viewed as OptEqs with some terms projected to infinite-dimensional spaces?
- Based on my understanding, the Patch Splitting appears to be essential for the proposed model. Since if there is not patch splitting then Eq.(6). would essentially be adding a scalar coefficient to the original update equation, which may not result in significant changes. Am I right?

---

> ### Author Rebuttal · Authors · 2023-08-10
>
> Thanks for your comments. The followings are our responses to your concerns.
>
> # 1. About the exponential term and the difference between our work and [1]
>
> In fact, the methodology and the motivation of [1] and our work are different:
>
> 1. [1]'s optimization problem is for one transformer layer while ours is for the whole equilibrium model.
> 2. [1]'s exponential term is applied for self-attention which only calculates the exponential of inputs' self inner product. However, ours is calculated for the difference between the input and our equilibrium model's output. Therefore, our GEQ is more similar to traditional attention modules like in SE-Net[2] and Graph Attention Networks[3] instead of self-attention models.
> 3. [1] propose the exponential term because they want to explain softmax in Transformer networks while we use Gaussian kernels to analyze equilibrium models with infinite width.
>
> Therefore, the formulations for our work and [1] are similar. They are two different works.
>
> # 2. About the stability of our GEQ in Equation (6).
>
> In our work, the exponential term in our GEQ is stable because the weights are constrained by weight normalization to ensure the convergence of Eqn(6):
>
> $\mathbf{z}^* = \sigma\left[e^{-\gamma\|\mathbf{U}\mathbf{x}+\mathbf{b}-\mathbf{Wz}^*\|^2_2} \mathbf{W}^\top(-\mathbf{Wz}^* +\mathbf{U}\mathbf{x}+\mathbf{b})\right]$
>
> Therefore, our GEQ's exponential term will be stable if the input is stable, which we use some normalization layers to ensure.
>
> # 3. About the feature extraction term.
>
> As we can see from OptEqs’s optimization problem:
>
> $\min_{\mathbf{z}} G(\mathbf{z};\mathbf{x}) = \min_{\mathbf{z}}\left[\mathbf{1}^\top
>   f(\mathbf{z})+ \frac{1}{2}\|\mathbf{z}\|_2^2-\left \langle \mathbf{U}\mathbf{x}+\mathbf{b},\mathbf{z} \right\rangle
> -\frac{1}{2}\|\mathbf{W}\mathbf{z}\|_2^2 \right].$
>
> The feature extraction term is $\langle \mathbf{Ux+b}, \mathbf{z} \rangle$, $\mathbf{x}$ here is input while $\mathbf{z}$ here is the output. Because when calculating the optimization problem's optimal condition to induce equilibrium models' architecture, it will produce the $\mathbf{Ux+b}$ term, which can be viewed as extracting useful features. Therefore, we call it the feature extraction term. We'll rewrite such a part to make it more clear in the future version.
>
> # 4. About "I wonder if we can say the expressive power of GEQ is strictly stronger than the original form of OptEqs".
>
> As our empirical results and theoretical analysis show,  GEQ’s performance is better than vanilla OptEqs with the proper choice of $\gamma$. However, if $\gamma$ is too large. For example, if $\gamma=\infty$, then the optimization problem for GEQ:
>
> $\min_{\mathbf{z}} G(\mathbf{z};\mathbf{x})=\min_{\mathbf{z}} \left[\mathbf{1}^\top f(\mathbf{z}) + \frac{1}{2}\|\mathbf{z}\|_2^2 - \frac{1}{2\gamma}e^{-\gamma\|\mathbf{U}\mathbf{x}+\mathbf{b}-\mathbf{Wz}\|^2_2}\right]$
>
> will become
>
> $\min_{\mathbf{z}} G(\mathbf{z};\mathbf{x})=\min_{\mathbf{z}} \left[\mathbf{1}^\top f(\mathbf{z}) + \frac{1}{2}\|\mathbf{z}\|_2^2\right]$
>
> then the final equilibrium state $\mathbf{z}$ will be $0$, which is a trivial solution.
>
> Thereby, we believe the expressive power of our GEQ is better than OptEqs with the proper choice of $\gamma$. Furthermore, although the choice of $\gamma$ is important, it won’t be hard to get a proper $\gamma$ for better performance. As we can see from the empirical section,  we choose one $\gamma$  for different datasets based on our analysis.
>
>
>
> # 5. About Patch Splitting and its necessity.
>
> Patch splitting is an important term and it is our GEQ's unique feature. Because even if we do patch splitting in OptEqs, its formulation still will be the same as the vanilla OptEqs:
>
> $\sum_{i=1}^N \langle (\mathbf{Ux+b})_i, \mathbf{z}_i\rangle = \langle (\mathbf{Ux+b}), \mathbf{z}\rangle,$
>
> where $\mathbf{z} = [\mathbf{z}_1,\mathbf{z}_2,...,\mathbf{z}_N]$. With the patch-splitting technique, our GEQ can concentrate on different parts based on their similarity like attentive modules.
>
> However, we need to clarify your comments on “if there is no patch splitting then Eq.(6). would essentially be adding a scalar coefficient to the original update equation, which may not result in significant changes.” Even without the patch splitting technique, our GEQ is still not the same as trivially scaling OptEqs because the scaling factor is related to $\mathbf{x}, \mathbf{z}$ and it is also learnable.
>
> # 6. About “In proposition 1, there’s a term $\sqrt{2\gamma}\Psi_\mathbf{W}^{(1)}$, what do the bolded 2, $\Phi$ and superscript (1) mean? I guess they shouldn’t be bolded and it is a typo?”
>
> They are typos and we’ll correct it in the future version.
>
> [1] Transformers from an Optimization Perspective
>
> [2] Squeeze-and-Excitation Networks
>
> [3] Graph Attention Networks

---

> ### Comment · Reviewer_p1Jk · 2023-08-15
>
> Thank the authors for the responce. Most of my concerns are well addressed, but I still have some questions.
>
> I agree that the exponential term in eq.(6) will not cause unstability. However, the author mentioned "the weights are constrained by weight normalization" in the rebutal. I am curious about how the weight normalization is implemented. I understand the weights need be normalized to ensure convergence of eq.(6), but I couldn't find anything in the Algorithm 1 (in appendix A.2) that ensures this constraint.
>
> Another point is, the author mentioned that even if without patch splitting, the proposed model is still essentially different from OptEqs because there is a learnable scaling factor which is related to $x$ and $z$. But I still don't understand why this scalar can lead to a essentially different solution. However complicated it is, it is just a scalar, isn't it? It would be helpful if authors can show some simple cases where adding a learnable scalar can lead to a essentially different fixed point.

---

> > ### Author Response · Authors · 2023-08-18
> >
> > Thanks for your reply. The followings are answers to your new questions:
> >
> > 1. About the weight normalization.
> >
> > We just use PyTorch weight normalization after the clarification of each convolution layer such as layer$\mathbf{W}$ and $\mathbf{U}$ like other equilibrium models[1,2]. It can be considered as rescaling  $\mathbf{W}$ by its norm after each update. Thereby, we forget to add it in Algorithm 1 as it is the forward process. Since it may lead to misunderstanding, we will clarify such a point in the future version.
> >
> > 2. About the “learnable scaler”.
> >
> > We think GEQ’s scaler is different from trivial scaling because it is a sample-dependent scaler. We think it may stabilize the original equilibrium model like OptEqs. For example, if we compared our GEQ with the following equilibrium model:
> >
> > $\mathbf{z} = \sigma(\mathbf{W}^\top(-\mathbf{Wz}+\mathbf{Ux+b}))$      (1)，
> >
> > it is the same as our GEQ without the exponential term and our GEQ can be formulated as:
> >
> > $\mathbf{z} = \sigma(e^{-\gamma\|-\mathbf{Wz}+\mathbf{Ux+b}\|_2^2}\mathbf{W}^\top(-\mathbf{Wz}+\mathbf{Ux+b}))$
> >
> > As one can see, our GEQ will scale the above equilibrium model’s output $\mathbf{z}$ for Eqn(1) when the difference between $\mathbf{Wz}$ and $\mathbf{Ux+b}$ is too large because the exponential term is small. However, when the difference is not too large, our GEQ’s output will be similar to the original equilibrium model’s outputs for Eqn(1) as the exponential term is around $1$. Thereby, it is different from scaling the equilibrium model with a fixed parameter.
> >
> > We assume that such a constraint can prevent some unstable behavior for the equilibrium models, owing to the higher controllability and stability inherent in the linear model $\mathbf{Ux}$. Such an assumption is also an intuitive motivation for our stability analysis.
> >
> > The empirical results also show that even without patch splitting, our GEQ can perform better than OptEqs as below on CIFAR-10:
> >
> > | Model | Model Size | Test Acc |
> > | --- | --- | --- |
> > | MOptEqs | $8$M | $94.6\%$ |
> > | GEQ(w/o patch) | $8$M | $94.9\%$ |
> > | GEQ | $8$M | $95.6\%$ |
> >
> > However, the advantages will be smaller than with the patch-splitting technique, which is consistent with your opinion.

---

> ### Comment · Reviewer_p1Jk · 2023-08-18
>
> Thank the authors for furthur clarification. All of my concerns are well addressed, so I would like to raise my rating to this paper to a clear accept accordingly. I encourage the authors to make it clear in the future revision that the weight matrices are normalized at each step of training, which is different from standard training process.

---

### Official Review · Reviewer_8r48 · 2023-07-07

**Soundness:** 3 good
**Presentation:** 3 good
**Contribution:** 3 good
**Rating:** 6
**Confidence:** 4

**Summary:**

In this paper, the authors propose to use Gaussian kernel in OptEqs (optimization-induced deep equilibrium models). Because of the more powerful capability of kernel, it can better capture the dynamic performance than linear models.


**Strengths:**

OptEqs is an interesting attempt to describe the training dynamics of DNN and hence making it more powerful (this paper's work) is interesting. (This does not mean that I appreciate the technique contribution of GEQ. To me, this is a natural incremental contribution from OptEqs.)

The reported performance is quite good, considering that GEQ implies a new model structure.


**Weaknesses:**

Using a Gaussian kernel instead of a linear is simple, which can also naturally lead to a better generalization bound. In other words, the discussion on the generalization bound is interesting but simply saying "tighter" is somehow trivial.

In my point of view, there is essential difference of applying GEQ to ImageNet-100 or the whole ImageNet.

I think that a new point of view of neural networks training itself is already interesting. It is not necessary to defeat standard neural networks. If this is the aim of this paper, the experiments should be enhanced. For example, ImageNet should be considered and
the setting (SGD with a step learning rate schedule) is fair but may be not sufficient. The following link gives the links of reported best training strategy: https://paperswithcode.com/sota/
CIFAR-10 with ResNet-18 is 95.55;
CIFAR-100 with ResNet-50-SAM is 85.2

In one word, I think the current experiments for evaluating GEQ are already good (when ImageNet is included). But it is far from giving the conclusion that GEQ is better than ResNet, etc.


My overall recommendation is currently positive, which is mainly based on numerical experiments. There are still many doubts about the performance. If the answer is not strong, I may lower my score. Also, I feel the theoretical and technique contributions of this paper is weak and hence will not fight for my current score, when other reviewers think this part should be improved.

**Questions:**

Replacing the linear operator in OptEqs by a nonlinear one is nice. But can the authors explain more why kernel is chosen? why not try e.g., MLP, of which the parameters could be trained.

Questions for numerical experiments could be found above, especially if the authors want to claim advantages over standard neural networks.

---

> ### Author Rebuttal · Authors · 2023-08-10
>
> Thanks for your comments. The followings are our responses to your concerns.
>
> # 1. About the contribution of GEQ.
>
> While OptEqs has already shown its ability in describing certain DNNs like ResNet, there are many limits on it:
>
> 1. Firstly, vanilla OptEqs can only model the behaviors of neural networks as they delve deeply into their architecture. However, its applicability is limited when attempting to analyze neural networks with infinite depth and width simultaneously.
> 2. Secondly, vanilla OptEqs is adept at describing fundamental neural networks characterized by linear layers (such as convolutions) with one pointwise non-linear activation module. Yet, it falls short when attempting to analyze network architectures with non-linear attentive modules.
> 3. Thirdly, the numerical performances for OptEqs are not satisfying.
>
> To solve the above problems, we decide to use involve Gaussian kernels in vanilla equilibrium models. The reason is mainly as below:
>
> 1. Using the Gaussian kernels can give us more insight into equilibrium models with infinite width since Gaussian kernels are usually related to infinite-dimensional space. Thereby, GEQ can not only be used to analyze deep neural models but also can give some insight into the training dynamics when the model’s depth and width increase both.
> 2. To extend equilibrium models for describing networks with some non-linear attentive modules, we decide to involve different non-linear kernels like polynomial, sigmoid, and Gaussian. And the empirical results show that the Gaussian kernel is the best. Thereby, we take a further step in exploring our GEQ’s theoretical and empirical advantages in this paper. As shown in Section 3.5, our GEQ can be viewed as an equilibrium model with attentive modules. Therefore, we may use it to analyze the training dynamics for neural networks with attention.
> 3. Our approach offers the potential to enhance both the generalization capability and stability of optimization-induced neural architectures like equilibrium models and provide valuable inspiration for researchers to devise more potent kernels for current models. Prior to our research, no exploration of new equilibrium models starts from our distinctive perspective.
>
> # 2. About numerical experiments.
>
> Firstly, we want to point out that the best result in https://paperswithcode.com/sota/ you report uses the auxiliary data. They won’t get such good performances if they train CIFAR models without auxiliary datasets.
>
> Secondly, we also finish the experiments for ResNet-50 with SAM on CIFAR-10 and CIFAR-100.
>
> ## CIFAR-10 with SAM:
>
> | Model | Model Size | Test Acc |  |
> | --- | --- | --- | --- |
> | ResNet-50 | $23$M | $95.5\pm0.4\%$ |  |
> | GEQ | $8$M | $\mathbf{95.9\pm0.3}\%$ |  |
>
> ## CIFAR-100 with SAM:
>
> | Model | Model Size | Test Acc |
> | --- | --- | --- |
> | ResNet-50 | $23$M | $78.4\pm0.3\%$ |
> | GEQ | $8$M | $\mathbf{78.9\pm0.2}\%$ |
>
> Our GEQ can outperform deep ResNets with less than half size even if changing optimizers to SAM.
>
> Furthermore, we also finish the experiments for our GEQ on ImageNet with SGD shown below:
>
> ## ImageNet:
>
> | Model | Model Size | Test Acc |
> | --- | --- | --- |
> | ResNet-18 | $13$M | $70.2\%$ |
> | ResNet-50 | $26$M | $75.1\%$ |
> | GEQ | $16$M | $\mathbf{75.9}\%$ |
>
> From the results, one can see that our GEQ can outperform deep ResNets with fewer parameters on larger ImageNet datasets.
>
> ## 3. About your concerns on “why kernel is chosen"？
>
> We use kernel as we deem that the poor performance for original equilibrium models is caused by the simple term $\langle \mathbf{Ux+b}, \mathbf{z} \rangle$ in OptEqs original optimization problem. Then inspired by traditional machine learning methods, we naturally choose to use kernel methods like Gaussian kernels. Apart from Gaussian, we also try other common kernels as shown in the Appendix A.1. We also list the formulation and results below.
>
> The formulation of equilibrium models with commonly used non-linear kernels:
>
> | Kernel | Hidden Optimization Problem | Equilibrium Model |
> | --------- | ---------- | -------------- |
> | Polynomial |  $\min_{\mathbf{z}}\left[\mathbf{1}^\top f(\mathbf{z})+ \frac{1}{2}\|\mathbf{z}\|_2^2-\left( \left\langle \mathbf{U}\mathbf{x}+\mathbf{b},\mathbf{z} \right\rangle\right)^d-\frac{1}{2}\|\mathbf{W}\mathbf{z}\|_2^2 \right]$  | $\mathbf{z}^* =  \sigma\left(\mathbf{W}^\top\mathbf{W}\mathbf{z}^*+ d\left( \left\langle \mathbf{U}\mathbf{x}+\mathbf{b},\mathbf{z} \right\rangle\right)^{d-1}\left(\mathbf{U}\mathbf{x}+\mathbf{b}\right)\right)$  |
> |Sigmoid | $\min_{\mathbf{z}}\left[\mathbf{1}^\top f(\mathbf{z})+ \frac{1}{2}\|\mathbf{z}\|_2^2-{\rm{tanh}}\left( \left\langle \mathbf{U}\mathbf{x}+\mathbf{b},\mathbf{z} \right\rangle\right) -\frac{1}{2}\|\mathbf{W}\mathbf{z}\|_2^2 \right]$ | $\mathbf{z}^* =  \sigma\left(\mathbf{W}^\top\mathbf{W}\mathbf{z}^*+ \left(1 - {\rm{tanh}}^2\left( \left\langle \mathbf{U}\mathbf{x}+\mathbf{b},\mathbf{z} \right\rangle\right)\right)\left(\mathbf{U}\mathbf{x}+\mathbf{b}\right)\right)$ |
> |Gaussian | $\min_{\mathbf{z}} \left[\mathbf{1}^\top f(\mathbf{z}) + \frac{1}{2}\|\mathbf{z}\|_2^2 - \frac{1}{2\gamma}e^{-\gamma\|\mathbf{U}\mathbf{x}+\mathbf{b}-\mathbf{Wz}\|^2_2}\right]$ | $\mathbf{z}^* = \sigma\left[e^{-\gamma\|\mathbf{U}\mathbf{x}+\mathbf{b}-\mathbf{Wz}^*\|^2_2}   \mathbf{W}^\top(-\mathbf{Wz}^* +\mathbf{U}\mathbf{x}+\mathbf{b})\right]$ |
>
> Their experiments are shown below, which demonstrate that GEQ is the best.
>
> | Model | Model Size | Accuracy |
> | --- | --- | --- |
> | MOptEqs | $8$M | $75.6\pm0.2\%$ |
> | MOptEqs (Polynomial) | $8$M | $75.1\pm0.4\%$ |
> | MOptEqs (Sigmoid) | $8$M | $76.1\pm0.3\%$ |
> | GEQ | $8$M | $\mathbf{78.2\pm0.2\%}$ |
>
> About MLP, if we use an MLP layer in the original $\langle \mathbf{Ux+b}, \mathbf{z} \rangle$ term to make it become $\langle \mathbf{Ux+b}, \mathbf{W}_{m}\mathbf{z} \rangle$.  Since it equals to $\langle \mathbf{W}_m^\top (\mathbf{Ux+b}), \mathbf{z} \rangle$, it will perform almost the same as vanilla OptEqs.

---

> > ### Comment · Reviewer_8r48 · 2023-08-18
> > **thanks**
> >
> > Thanks for the additional results. I also read the discussions with other reviewers.
> >
> > For my question, still I am not well convinced why there is significant improvement by simply using a kernel (the discussion about MLP is incorrect, since MLP is not a linear mapping when there is nonlinear activation function). So I would like to keep my score unchanged.

---

> > > ### Author Response · Authors · 2023-08-18
> > >
> > > Thanks for your reply. We would like to correct the answers about MLP since we neglect MLP’s non-linear layer in the former answer:
> > >
> > > 1. We consider kernels instead of MLP because of the following reasons:
> > >
> > >     1.  From the view of the equilibrium model’s hidden optimization problem, using MLP will make it hard to obtain the equilibrium model’s formulation.
> > >
> > >         For example, if we adopt MLP (denote as $g$) in the equilibrium model’s hidden optimization problem:
> > >
> > >         $\min_{\mathbf{z}} G(\mathbf{z};\mathbf{x}) = \min_{\mathbf{z}}\left[\mathbf{1}^\top
> > >           f(\mathbf{z})+ \frac{1}{2}\|\mathbf{z}\|_2^2-\left \langle \mathbf{U}\mathbf{x}+\mathbf{b},g(\mathbf{z}) \right\rangle
> > >         -\frac{1}{2}\|\mathbf{W}\mathbf{z}\|_2^2 \right].$
> > >
> > >         When $z$ is a scaler, then it’s somehow equivalent to our GEQ, since it will also induce an attention scaler $g'(z)$.
> > >
> > >         However, $\nabla g(\mathbf{z})$ is a matrix when $\mathbf{z}$  is a vector in most common cases. Therefore, the formulation of equilibrium models will be complicated. And its convergence may be also hard to constrain because of its formulation.
> > >
> > >     2.  Apart from its complicated formulation, another important reason why we use kernels instead of MLPs is that kernels can provide us with much more theoretical insights than MLPs.  For example, gaussian kernels can inspire us that our new model may be more stable and may also enable us to analyze the performance of wider models.
> > >
> > > 2. We also want to clarify that simply adding non-linear activations in equilibrium model’s equilibrium function can not make it perform much better. For example,  MDEQ has several non-linear activation layers in its equilibrium equation since they adopt a residual block in their architecture. However, it does not make its performance much better compared with OptEqs and MOptEqs.  What’s worse, it makes MDEQ lose its ability to be interpreted by an optimization problem.
> > >
> > >     We think the reason is the equilibrium model is already a kind of deep neural network with non-linear layers. Therefore simply adding nonlinear activations inside its equilibrium function will not help much. Thus we don’t think the number of non-linear layers in MOptEqs or OptEqs’s equilibrium equation is the key reason for their weak performances.
> > >
> > > 3. Comparing our GEQ’s architecture (Figure 1) with other equilibrium models, we may give one intuitive reason for our better performance against other equilibrium models: our GEQ is similar to equilibrium models with attention modules, which is the first trial in equilibrium models especially from the optimization view.
> > >
> > >     And we also want to note that such a difference is also a contribution of our model compared with former works:
> > >
> > >     1. Firstly, we can explain why attention models are better. Former vanilla models like ResNet and MLP can be viewed as vanilla OptEqs with linear kernels while neural networks with attention modules can be viewed as equilibrium models with non-linear kernels. Since the expressive power for non-linear kernels is better than linear ones. The performance of neural networks with attention is better.
> > >
> > >     2. Secondly, like ResNets and MLP’s training dynamics can be analyzed by vanilla OptEqs. Our GEQ may be used to analyze the training dynamics of attention networks and inspire new designs on attention modules by findings new non-linear kernels. We will further explore the properties and new architectures of attention modules from such a view.

---

### Comment · Area_Chair_XgmB · 2023-08-10
**Bounds tightness**

Dear authors,

A concern that the reviewers have not raised but that I think is an important point is the tightness of the bounds in prop.2,3 and 4.

For instance, take prop.3. It only shows that the Lipschitz constant of $f_{geq}$, defined as $\sup_{x, y} \frac{\|f_{geq}(x) - f_{geq}(y)\|}{\|x - y\|}$ is **upper-bounded** by $L_{deq}$. Similarly, you show that the Lipschitz constant of $f_{opteq}$ is upper-bounded by $L_{opteq}$. Then, you give conditions under which $L_{deq}<L_{opteq}$, and then conclude that "our GEQ’s outputs are more stable under perturbations". This seems like a mathematical fallacy to me, since your sentence seems to imply that the Lipschtiz constant of $f_{deq}$ is smaller than that of $f_{opteq}$. This fact is never proven anywhere and is not discussed in the paper. In order to have such a strong statement, you need to prove that the bounds of these propositions are tight.

Can you comment on this point?

Thanks a lot !

AC

---

> ### Author Response · Authors · 2023-08-10
>
> Thanks for your comments. The followings are the response to your concerns.
>
> In our paper, we are trying to compare the worst-case performance for OptEqs and GEQ. And such a bound is tight since we can find some cases in which the upper bound can be achieved. For example, in Proposition 3, the upper bound can be achieved if $\mathbf{W}=\mu \mathbf{I}, \mathbf{U}=\mu \mathbf{I}$ and other parameterizations may also hold the equality. Although such a parameterization may be somehow extraordinary, it can still happen with certain data during training. Therefore, both our work and former works (like [1]) try to estimate the worst cases since we don’t set many assumptions in our analysis.
>
> We need to point out that such an analysis can also guide empirical studies, as our empirical results also demonstrate the generalization performance and stability of our GEQ compared with the vanilla OptEqs.
>
> [1] Estimating Lipschitz constants of monotone deep equilibrium models. ICLR 2021.

---

> > ### Comment · Area_Chair_XgmB · 2023-08-11
> >
> > Dear authors,
> >
> > Thanks for your response.
> >
> > If you take a general set of parameters $U$, $W$, can you find $x_1$ and $x_2$ such that the bounds in prop.3. are tight? If not, how do you quantify the tightness of your bounds?
> >
> > Best,
> > AC

---

> > > ### Author Response · Authors · 2023-08-11
> > >
> > > Dear Aera Chair,
> > >
> > > Thanks for your comments and the following are our responses.
> > >
> > > The equality cannot always hold for any $\mathbf{W}$ and $\mathbf{U}$. Therefore, the upper bound we calculate is not the Lipschitz constant for every $\mathbf{W}$ and $\mathbf{U}$.  For example, if $\mathbf{Ux}$ lies in $\mathbf{W}^\top\mathbf{W}$’s non-space and $\mathbf{Ux}>0$ for any $\mathbf{x}$ in the dataset. Then $z_x$ will be zero and therefore $\|z_{x_1} -z_{x_2}\|_2$ will always be zero.
> > >
> > > However, we think our comparison is still meaningful. Since we not only consider the worst case of the input but also the worst weights since you cannot ensure the parameters can be obtained as you wish. In this scenario, the upper bound is tight. Therefore, we can compare the worst cases of our GEQ and OptEqs and then conclude that our architecture is better.  For example, a more general case is that if $\mathbf{W}^\top\mathbf{W}$ and $\mathbf{U}'s$ eigenspaces for top eigenvalues align together. And $\mathbf{z}_x = (\mathbf{I}+\mathbf{W}^\top\mathbf{W})^{-1}\mathbf{U}\mathbf{x}$ is OptEqs output if $(\mathbf{I}+\mathbf{W}^\top\mathbf{W})^{-1}\mathbf{U}\mathbf{x} > 0$. Then for $\mathbf{x}_1,\mathbf{x}_2$ lie in $\mathbf{U}$’s eigenspace for top eigenvalue $\mu$. Their difference is just $\frac{\mu}{1-\mu^2}\|\mathbf{x}_1 - \mathbf{x}_2\|_2$.
> > >
> > > Since you cannot ensure that the model can be trained as you wish and the modification in data may also lead the whole model trained to be some vulnerable cases (e.g. backdoor attack[1]), we think comparing the worst cases for both weights and inputs are necessary and more realistic. In other words, we are trying to compare the stability of the worst GEQ and worst OptEqs.
> > >
> > > [1] Backdoor learning: A survey
> > >
> > > Best wishes!

---

### Decision · Program_Chairs · 2023-09-21

**Decision:**

Accept (poster)

**Comment:**

This paper proposed a new equilibrium model using kernels. The theoretical motivation behind the proposal is quite clear, as the authors demonstrate tighter generalization bounds and smoother layers. The experiments then demonstrate clearly the merits of the approach on several tasks.

The reviewers were generally positive about this work, and the authors adequately answered their concerns. Reviewer CdMn still scores 4, but this is mostly based on minor mathematical misunderstandings that the authors have clarified and not on some concern regarding the method/soundness/evaluation.

 In the final version of the paper, the authors must be clearer about their claims in 3.3 - 3.4: these upper bounds are not tight (this should be explicit!); hence it is a fallacy to claim that "Therefore, our GEQ can show better classification performance on the test set" or "Therefore, we can conclude that our GEQ’s outputs are more stable under perturbations.", this needs to be mitigated as the authors did in discussion with the AC.